# On Output Activation Functions for Adversarial Losses: A Theoretical Analysis via Variational Divergence Minimization and An Empirical Study on MNIST Classification

## Abstract

Recent years have seen adversarial losses been applied to many fields. Their applications extend beyond the originally proposed generative modeling to conditional generative and discriminative settings. While prior work has proposed various output activation functions and regularization approaches, some open questions still remain unanswered. In this paper, we aim to study the following two research questions: 1) What types of output activation functions form a well-behaved adversarial loss? 2) How different combinations of output activation functions and regularization approaches perform empirically against one another? To answer the first question, we adopt the perspective of variational divergence minimization and consider an adversarial loss well-behaved if it behaves as a divergence-like measure between the data and model distributions. Using a generalized formulation for adversarial losses, we derive the necessary and sufficient conditions of a well-behaved adversarial loss. Our analysis reveals a large class of theoretically valid adversarial losses. For the second question, we propose a simple comparative framework for adversarial losses using discriminative adversarial networks. The proposed framework allows us to efficiently evaluate adversarial losses using a standard evaluation metric such as the classification accuracy. With the proposed framework, we evaluate a comprehensive set of 168 combinations of twelve output activation functions and fourteen regularization approaches on the handwritten digit classification problem to decouple their effects. Our empirical findings suggest that there is no single winning combination of output activation functions and regularization approaches across all settings. Our theoretical and empirical results may together serve as a reference for choosing or designing adversarial losses in future research.

## 1 Introduction

Generative adversarial networks (GANs) (Goodfellow et al., 2014) are a class of unsupervised machine learning algorithms. The core of a GAN is an *adversarial loss* that learns to measure the discrepancy between the real data distribution and the distribution of the generated samples. While adversarial losses are originally used to train a generative model (Goodfellow et al., 2014), their applications extend beyond unsupervised generative modeling to other supervised tasks such as conditional generative modeling (Mirza & Osindero, 2014) and discriminative modeling (dos Santos et al., 2017). Moreover, prior work has proposed a range of adversarial losses in different forms, motivated and derived from various existing divergences or distances. For example, the classic minimax GANs (Goodfellow et al., 2014) is motivated from the Jensen–Shannon divergence, Wasserstein GANs (Arjovsky et al., 2017) from the Wasserstein distance, least squares GANs (Mao et al., 2017) from the Pearson $\chi^2$ divergence, and $f$-GANs (Nowozin et al., 2016) from the $f$-divergence. Notably, these GAN formulations adopt different output activation functions in their training objectives. However, it remains unclear what types of output activation functions form a well-behaved adversarial loss. Moreover, apart from the output activation functions, the adversarial losses proposed in prior work differ also in the regularization approach adopted. Some representative examples include the gradient penalties (Gulrajani et al., 2017; Kodali et al., 2017; Mescheder et al., 2018) and the spectral normalization (Miyato

et al., 2018). However, it remains unclear how the output activation functions and the regularization approach contribute respectively to the empirical performance of an adversarial loss.

In this paper, we aim to study the following two research questions:

**Question 1.** *What types of output activation functions form a well-behaved adversarial loss?*

**Question 2.** *How different combinations of output activation functions and regularization approaches perform empirically against one another?*

Our contributions to Question 1 are based on the *variational divergence minimization* perspective proposed by Nowozin et al. (2016) and the concept of *adversarial divergence* proposed by Liu et al. (2017). Specifically, we consider an adversarial loss *well-behaved* if it behaves as a divergence-like measure between the data and model distributions.[1] Using a generalized formulation proposed by Jolicoeur-Martineau (2019), we derive the necessary and sufficient conditions for an adversarial loss to be well-behaved. Our theoretical analysis reveals a large class of theoretically valid adversarial losses. Based on our findings, we propose two new adversarial losses—the *absolute* loss and the *asymmetric* loss. The absolute loss features a mixture of characteristics from two existing adversarial losses, while the asymmetric loss possesses distinct characteristics from other adversarial losses. While both losses do not originate from any underlying divergence, we will show in our empirical study that they can achieve similar performance against other adversarial losses.

For Question 2, some attempts have been made towards decoupling the effects of output activation functions and regularization approaches. For example, Fedus et al. (2018) showed that the original nonsaturating GAN (Goodfellow et al., 2014) can achieve comparable performance as the WGAN-GP model (Gulrajani et al., 2017) when using the same gradient penalty. However, due to the high computational cost for conducting a large-scale comparative study, Fedus et al. (2018); Lucic et al. (2018); Kurach et al. (2019) evaluate only five, seven and ten, respectively, combinations of output activation functions and regularization approaches. This necessitates the needs for an efficient way to compare adversarial losses. In view of the various tasks that adversarial losses are applied to, we propose to adopt the discriminative adversarial networks (DANs) for our comparative study as we can efficiently evaluate different adversarial losses using a standard evaluation metric such as the classification accuracy. For simplicity and computation cost concerns, we consider the MNIST handwritten digit classification problem (LeCun et al., 1998). With the proposed framework, we systematically evaluate 168 adversarial losses featuring the combinations of twelve output activation functions and fourteen regularization approaches. Moreover, we empirically study the effects of the Lipschitz constant (Arjovsky et al., 2017), penalty weights (Mescheder et al., 2018), momentum terms (Kingma & Ba, 2014). We also study the robustness of different regularization approaches to data imbalance.

Our findings suggest that *there is no single winning combination of output activation functions and regularization approaches across all settings.* The global lowest error rate is achieved by combining the relativistic hinge loss with the two-side local gradient penalty. Moreover, by comparing the minimum error rate achieved for each output activation function across different regularization approaches, we find that the hinge loss significantly outperforms the classic loss. Similarly, the coupled and local two-side gradient penalties significantly outperform other gradient penalties, no matter whether the spectral normalization is applied or not. Through comparing the maximum error rate achieved for each output activation function across different regularization approaches, we find that our proposed asymmetric loss is the most robust output activation function. Similarly, the spectral normalization improves the robustness for all gradient penalties. Although it is unclear whether the empirical results we obtain for DANs can generalize to unconditional or conditional GANs,[2] we argue that the idea of adversarial losses is fairly generic and they can be used to train either generative or discriminative models. Hence, our empirical results still contribute to the goal towards a deeper understanding of adversarial losses.

---

[1]The model distribution is the distribution of the generated samples. It is implicitly defined by the generator (see Section 3.1).
[2]We note that even though the generator is no longer a generative model in a DAN, the discriminator is still trained to minimize the same loss function as that in a conditional GAN (see Section 3.7).

## 2  Related Work

Prior work has studied the working of adversarial losses from different perspectives. Some view adversarial losses from the perspective of variational divergence minimization (Nowozin et al., 2016). From this view, Nowozin et al. (2016) derived the $f$-GAN family based on the $f$-divergences. However, the output activation functions used for the data and model distribution need to be the conjugate function of each other in their formulations. In our theoretical analysis, we will relax this restriction and work on a more generalized formulation of adversarial losses. Further, Liu et al. (2017) proposed a broader concept of adversarial divergence and showed that if the objective function is an adversarial divergence with some conditions, then using a restricted discriminator family has a moment-matching effect. However, they did not show what output activation functions lead to a valid adversarial divergence. In this paper, we will derive the necessary and sufficient conditions for a well-behaved adversarial loss.

Some view adversarial losses from the perspective of integral probability metric view (Arjovsky & Bottou, 2017; Mroueh et al., 2017; Li et al., 2017; Mroueh & Sercu, 2017). In particular, Arjovsky & Bottou (2017) showed that by imposing a Lipschitz constraint to the discriminator, the training objective for the generator corresponds to minimizing the Wasserstein distance between the data and model distributions. In this formulation, the output activation functions used for the data and model distribution need to be the same. Later, some follow-up work proposed to enforce the Lipschitz constraint using gradient penalties (Gulrajani et al., 2017; Kodali et al., 2017; Wei et al., 2018; Mescheder et al., 2018) and spectral normalization (Miyato et al., 2018). Moreover, Fedus et al. (2018) suggested that these regularization approaches extend beyond the integral probability metric view and can be applied to other GAN formulations. Motivated by this observation, we will examine in our empirical study combining different output activation functions and regularization approaches to decouple their effects. Moreover, they suggested that the training dynamic of a GAN is closer to approaching a Nash equilibrium of the adversarial game than minimizing a specific divergence. Nonetheless, we can view an adversarial loss as a dynamic, divergence-like measure that quantifies the discrepancy between the data and model distributions. This view will serve as the foundation of our theoretical analysis.

Another relevant line of research are the comparative studies of adversarial losses. Fedus et al. (2018) showed that both the gradient penalty (Gulrajani et al., 2017) and spectral normalization (Miyato et al., 2018) can also improve the performance for a nonsaturating GAN (Goodfellow et al., 2014) in addition to the Wasserstein GAN (Arjovsky et al., 2017) they are originally proposed for. Later, Lucic et al. (2018) conducted a large-scale empirical study and showed that the seven GAN models considered in their study can reach similar scores given enough computation budgets. Similarly, Kurach et al. (2019) showed that the spectral normalization (Miyato et al., 2018) consistently improves the sample quality for the four output activation functions considered, while the gradient penalty (Gulrajani et al., 2017) require a large computation budget to find a good hyperparameter setting. However, due to the high computational cost for conducting a large-scale experiment, Fedus et al. (2018); Lucic et al. (2018); Kurach et al. (2019) examine only five, seven and ten, respectively, combinations of output activation functions and regularization approaches. In this paper, we will propose a simple comparative framework for efficiently comparing adversarial losses based on a discriminative adversarial network. With the proposed framework, we evaluate a comprehensive set of 168 combinations of twelve output activation functions and fourteen regularization approaches.

## 3  Background

In this section, we provide key background knowledge useful for our discussions in the rest of the paper.

### 3.1  Generative adversarial networks

A generative adversarial network (GAN) (Goodfellow et al., 2014) is a generative latent variable model that aims to learn a generative model $G : \mathcal{Z} \mapsto \mathcal{X}$ that maps some latent space $\mathcal{Z}$ to the data space $\mathcal{X}$. A discriminative model $D : \mathcal{X} \mapsto \mathbb{R}$ defined on $\mathcal{X}$ is trained alongside $G$ to provide guidance for $G$. Now, let $P_d$ denote the data distribution and $P_z$ the latent distribution. The original GAN is formulated as a two-player

minimax game between the generator $G$ and discriminator $D$ as follows.

$$\min_G \max_D \mathbb{E}_{\boldsymbol{x} \sim P_d}[\log D(\boldsymbol{x})] + \mathbb{E}_{\boldsymbol{z} \sim P_z}[\log(1 - D(G(\boldsymbol{z})))]. \tag{1}$$

Alternatively, let $P_g$ be the *model distribution* implicitly defined by $G(\boldsymbol{z})$ with $\boldsymbol{z} \sim P_z$.

$$\min_G \max_D \mathbb{E}_{\boldsymbol{x} \sim P_d}[\log D(\boldsymbol{x})] + \mathbb{E}_{\tilde{\boldsymbol{x}} \sim P_g}[\log(1 - D(\tilde{\boldsymbol{x}}))]. \tag{2}$$

We will refer to this original formulation to the classic minimax GAN for the rest of this paper.

## 3.2 A variational divergence minimization view of adversarial losses

Some prior work investigated adversarial losses from the perspective of variational divergence minimization. Nowozin et al. (2016) proposed to minimize the following variational lower bound of a $f$-divergence derived by Nguyen et al. (2010):

$$D_f(P_d, P_g) \geq \sup_{D \in \mathcal{D}} \mathbb{E}_{\boldsymbol{x} \sim P_d}[D(\boldsymbol{x})] - \mathbb{E}_{\tilde{\boldsymbol{x}} \sim P_g}[f^*(D(\tilde{\boldsymbol{x}}))], \tag{3}$$

where $\mathcal{D}$ is an arbitrary set of functions $D : \mathcal{X} \mapsto \mathbb{R}$ and $f^*$ is the corresponding conjugate function for an $f$-divergence. Nowozin et al. (2016) showed that this can be formulated as an adversarial minimax game similar to a GAN:

$$\min_G \max_D \mathbb{E}_{\boldsymbol{x} \sim P_d}[g_f(D(\boldsymbol{x}))] + \mathbb{E}_{\tilde{\boldsymbol{x}} \sim P_g}[-f^*(g_f(D(\tilde{\boldsymbol{x}})))], \tag{4}$$

where $g_f : \mathbb{R} \mapsto dom(f^*)$ is an output activation function specific to the $f$-divergence used. The $f$-GAN family given by Equation (4) includes some representative GAN formulations, including the classic minimax GANs (Goodfellow et al., 2014) and least-square GANs (Mao et al., 2017).

## 3.3 An integral probability metric view of adversarial losses

Some prior work has investigated viewing an adversarial loss as an estimator for an integral probability metric (IPM) (Müller, 1997). Arjovsky et al. (2017) considered the Wasserstein distance and proposed to minimize the following dual form given by the Kantorovich–Rubinstein duality:

$$W(P_d, P_g) = \sup_{||f||_L \leq 1} \mathbb{E}_{\boldsymbol{x} \sim P_d}[f(\boldsymbol{x})] - \mathbb{E}_{\tilde{\boldsymbol{x}} \sim P_g}[f(\tilde{\boldsymbol{x}})], \tag{5}$$

Arjovsky et al. (2017) then proposed the following Wasserstein GAN formulation:

$$\min_G \max_{D \in \mathcal{D}} \mathbb{E}_{\boldsymbol{x} \sim P_d}[f(\boldsymbol{x})] - \mathbb{E}_{\tilde{\boldsymbol{x}} \sim P_g}[f(\tilde{\boldsymbol{x}})], \tag{6}$$

where $\mathcal{D}$ is the set of all 1-Lipschitz functions. Other examples of IPM-based GANs include McGANs (Mroueh et al., 2017), MMD GANs (Li et al., 2017) and Fisher GANs (Mroueh & Sercu, 2017).

## 3.4 Relativistic GANs

Jolicoeur-Martineau (2019) argued that the generator $G$ should also be trained to decrease the probability that the real samples are classified as real by the discriminator $D$. Based on this argument, they proposed two variants for GANs called the relativistic GANs and the relativistic average GANs. In this paper, we consider the relativistic average GAN as follows.

$$\min_D \; -\mathbb{E}_{\boldsymbol{x} \sim P_d}\left[\log \hat{D}(\boldsymbol{x})\right] - \mathbb{E}_{\tilde{\boldsymbol{x}} \sim P_g}\left[\log\left(1 - \hat{D}(\tilde{\boldsymbol{x}})\right)\right], \tag{7}$$

$$\min_G \; -\mathbb{E}_{\tilde{\boldsymbol{x}} \sim P_g}\left[\log \hat{D}(\tilde{\boldsymbol{x}})\right] - \mathbb{E}_{\boldsymbol{x} \sim P_d}\left[\log\left(1 - \hat{D}(\boldsymbol{x})\right)\right], \tag{8}$$

Table 1: Distribution $p_{\hat{x}}$ and function $R$ in Equation (11) for common gradient penalties, where $k \in \mathbb{R}$ is the Lipschitz constant and $c \in \mathbb{R}$ is a hyperparameter.

|  | $p_{\hat{x}}$ | $R(r)$ |
|---|---|---|
| Two-side coupled gradient penalty (Gulrajani et al., 2017) | $P_d + U[0,1]\,(P_g - P_d)$ | $(r-k)^2$ |
| One-side coupled gradient penalty (Gulrajani et al., 2017) | $P_d + U[0,1]\,(P_g - P_d)$ | $\max(r,k)$ |
| Two-side local gradient penalty (Kodali et al., 2017) | $P_d + c\,N(0,I)$ | $(r-k)^2$ |
| One-side local gradient penalty (Kodali et al., 2017) | $P_d + c\,N(0,I)$ | $\max(r,k)$ |
| $R_1$ gradient penalty (Mescheder et al., 2018) | $P_d$ | $r^2/2$ |
| $R_2$ gradient penalty (Mescheder et al., 2018) | $P_g$ | $r^2/2$ |

where $\hat{D}(\boldsymbol{x}) = \sigma\big(D(\boldsymbol{x}) - \mathbb{E}_{\tilde{\boldsymbol{x}} \sim P_g}\left[D(\tilde{\boldsymbol{x}})\right]\big)$, $\hat{D}(\tilde{\boldsymbol{x}}) = \sigma\big(D(\tilde{\boldsymbol{x}}) - \mathbb{E}_{\boldsymbol{x} \sim P_d}\left[D(\boldsymbol{x})\right]\big)$, $\sigma(\cdot)$ is the sigmoid function. We also consider the relativistic average hinge GAN as follows.

$$\min_{D} \; -\mathbb{E}_{\boldsymbol{x} \sim P_d}\left[\max\left(0, 1 - \hat{D}(\boldsymbol{x})\right)\right] - \mathbb{E}_{\tilde{\boldsymbol{x}} \sim P_g}\left[\max\left(0, 1 + \hat{D}(\tilde{\boldsymbol{x}})\right)\right], \tag{9}$$

$$\min_{G} \; -\mathbb{E}_{\tilde{\boldsymbol{x}} \sim P_g}\left[\max\left(0, 1 - \hat{D}(\tilde{\boldsymbol{x}})\right)\right] - \mathbb{E}_{\boldsymbol{x} \sim P_d}\left[\max\left(0, 1 + \hat{D}(\boldsymbol{x})\right)\right]. \tag{10}$$

### 3.5 Gradient penalties

Originally proposed in (Gulrajani et al., 2017), a gradient penalty is a regularization approach to impose a soft Lipschitz constraint on the discriminator in a Wasserstein GAN (Arjovsky et al., 2017). Prior work has proposed different variants of gradient penalties (Kodali et al., 2017; Mescheder et al., 2018). In general, they take the following generalized form as a regularization term added to the discriminator loss function:

$$\lambda\,\mathbb{E}_{\hat{\boldsymbol{x}} \sim p_{\hat{x}}}[R(||\nabla_{\hat{\boldsymbol{x}}} D(\hat{\boldsymbol{x}})||)], \tag{11}$$

where $\lambda \in \mathbb{R}$ is a hyperparameter, which we will refer to as the *penalty weight*, and $R : \mathbb{R} \mapsto \mathbb{R}$ is a real function. The distribution $p_{\hat{x}}$ defines where the gradient penalty is enforced. This formulation covers some representative gradient penalties proposed in prior work, including the coupled gradient penalty used in the WGAN-GP model (Gulrajani et al., 2017), the local gradient penalty used in the DRAGAN model (Kodali et al., 2017), and the $R_1$ and $R_2$ gradient penalties (Mescheder et al., 2018), as summarized in Table 1. In addition, Figure 1 illustrates the support of $p_{\hat{x}}$, i.e., where gradient penalty is imposed.

### 3.6 Spectral normalization

Another regularization approach for Wasserstein GANs (Arjovsky et al., 2017) is the spectral normalization proposed by Miyato et al. (2018). By normalizing the spectral norm of the weight matrix for each layer of the discriminator, the spectral normalization imposes a hard Lipschitz constraint on the discriminator. As compared to a gradient penalty, which imposes a local regularization as illustrated in Figure 1, the spectral normalization imposes a global regularization for the discriminator.

### 3.7 Discriminative adversarial networks

A discriminative adversarial network (dos Santos et al., 2017) is essentially a conditional GAN (Mirza & Osindero, 2014) where both the generator and discriminator are discriminative models. In a DAN, the generator aims to predict the label of a real data sample, whereas the discriminator takes as input either a real pair—"(real data, real label)" or a fake pair—"(real data, fake label)", and aims to examine its authenticity. Mathematically, the generator in a DAN learns the mapping from the data space to the label space, i.e., $\mathcal{X} \mapsto \mathcal{Y}$, where $\mathcal{Y}$ denotes the label space. This is different from a conditional GAN, where the generator learns a mapping from the label space to the data space, i.e., $\mathcal{Y} \mapsto \mathcal{X}$. However, the discriminator in either a DAN or a conditional GAN learns the same mapping $\mathcal{X} \times \mathcal{Y} \mapsto \mathbb{R}$.

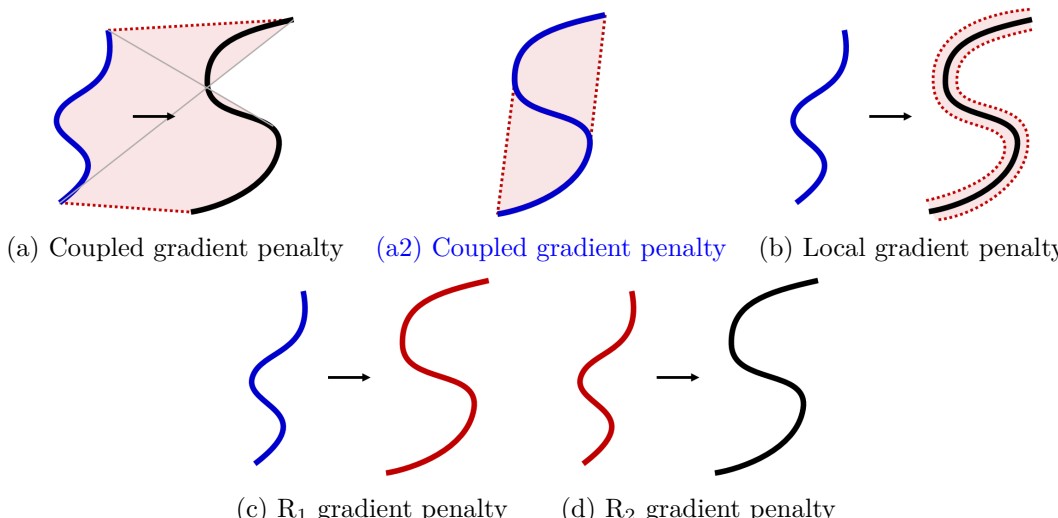

(a) Coupled gradient penalty  (a2) Coupled gradient penalty  (b) Local gradient penalty

(c) $R_1$ gradient penalty      (d) $R_2$ gradient penalty

Figure 1: Illustrations of the support of $p_{\tilde{x}}$, i.e., the region where the gradient penalty is imposed, for the gradient penalties considered in this paper, shown as the red shaded area in (a) and (b) and the red curves in (c) and (d). The blue and black curves denote the model and data manifolds, respectively. (a2) shows the case when the generator perfectly fabricates the data distribution, i.e., $P_g = P_d$. For (c) and (d), the gradient penalty is enforced directly on the model and data manifolds, respectively.

## 4  Theoretical Results

In this section, we present our theoretical results on Question 1: What types of output activation functions form a well-behaved adversarial loss? For the rest of the paper, we will adopt the following notations: $D$ denotes the discriminator and $G$ denotes the generator in a GAN. $P_d$ denotes the data distribution defined on the data space $\mathcal{X}$. $P_z$ denotes the latent distribution defined on the latent space $\mathcal{Z}$. $P_g$ denotes the *model distribution* implicitly defined by $G(\boldsymbol{z})$ with $\boldsymbol{z} \sim p_z$.

### 4.1  Generalized formulation of adversarial losses

Following Jolicoeur-Martineau (2019), we adopt a generalized form of adversarial losses as follows:

$$\max_D \mathbb{E}_{\boldsymbol{x} \sim P_d}[f(D(\boldsymbol{x}))] + \mathbb{E}_{\tilde{\boldsymbol{x}} \sim P_g}[g(D(\tilde{\boldsymbol{x}}))], \tag{12}$$

$$\min_G \mathbb{E}_{\tilde{\boldsymbol{x}} \sim P_g}[h(D(\tilde{\boldsymbol{x}}))], \tag{13}$$

where $f$, $g$ and $h$ are all real functions, i.e., $\mathbb{R} \mapsto \mathbb{R}$. We will refer to $f$, $g$ and $h$ as the *output activation functions*. This generalized formulation covers a wide range of adversarial losses proposed in prior work, including the classic minimax and nonsaturating GANs (Goodfellow et al., 2014), Wasserstein GANs (Arjovsky et al., 2017), least squares GANs (Mao et al., 2017) and hinge GANs (Lim & Ye, 2017; Tran et al., 2017), as summarized in Table 2.[3]

In the following theoretical analysis, we will focus only on the output activation functions $f$ and $g$. Specifically, we consider the minimax formulation where $h = g$:

$$\min_G \max_D \mathbb{E}_{\boldsymbol{x} \sim P_d}[f(D(\boldsymbol{x}))] + \mathbb{E}_{\tilde{\boldsymbol{x}} \sim P_g}[g(D(\tilde{\boldsymbol{x}}))]. \tag{14}$$

We note that from the perspective of either variational divergence minimization or integral probability metric, the generator should minimize the divergence-like measure estimated by the discriminator, and accordingly we have $h = g$. However, some prior work has investigated setting $h$ different from $g$. For example, the

---

[3]Notably, this formulation does not cover the relativistic GANs proposed by Jolicoeur-Martineau (2019) (see Section 3.4), which we will include in our empirical comparative study in Section 6.

Table 2: Output activation functions for the adversarial losses discussed in this paper (see Equation (12) and Equation (13) for the definitions of $f$, $g$ and $h$). In the rightmost column, $y^*$ denotes the common root of $f(y) = g(y)$ and $f'(y) = -g'(y) \neq 0$ (see Section 4.4 for more details).

| | $f$ | $g$ | $h$ | $y^*$ |
|---|---|---|---|---|
| Classic minimax (2014) | $-\log(1 + e^{-y})$ | $-y - \log(1 + e^{-y})$ | $-y - \log(1 + e^{-y})$ | 0 |
| Classic nonsaturating (2014) | $-\log(1 + e^{-y})$ | $-y - \log(1 + e^{-y})$ | $\log(1 + e^{-y})$ | 0 |
| Wasserstein (2017) | $y$ | $-y$ | $-y$ | 0 |
| Least squares (2017) | $-(1 - y)^2$ | $-y^2$ | $(1 - y)^2$ | 1/2 |
| Hinge (2017; 2017) | $\min(0, y - 1)$ | $\min(0, -1 - y)$ | $-y$ | 0 |
| Absolute | $-|1 - y|$ | $-|y|$ | $|1 - y|$ | 1/2 |
| Asymmetric | $-|y|$ | $-y$ | $-y$ | 0 |

classic nonsaturating loss proposed by Goodfellow et al. (2014) has a different $h$ with opposite concavity as compared to the classic minimax loss (see Table 2). While our theoretical analysis concerns with only $f$ and $g$, we will empirically examine the effects of the output activation function $h$ in Section 6.2.

## 4.2 Desirable properties for adversarial losses

Following the perspective of variational divergence minimization, we can see that if the discriminator is able to reach optimality, the training objective for the generator is

$$L_G = \max_D \; \mathbb{E}_{\boldsymbol{x} \sim P_d}[f(D(\boldsymbol{x}))] + \mathbb{E}_{\tilde{\boldsymbol{x}} \sim P_g}[g(D(\tilde{\boldsymbol{x}}))] \,. \tag{15}$$

In principle, the discriminator in this formulation is responsible for providing a measure of the discrepancy between $P_d$ and $P_g$, which will then serve as the training objective for the generator to push $P_g$ towards $P_d$. Hence, we would like such an adversarial loss to be a divergence-like measure between $P_g$ and $P_d$. Following this view of adversarial divergence (Liu et al., 2017), we can define the following two properties of a well-behaved adversarial losses.

**Property 1** (Weak desired property of a adversarial loss)**.** *For any data distribution $P_d$, the generator loss $L_G$ has a global minimum at $P_g = P_d$ with respect to the model distribution $P_g$.*

**Property 2** (Strong desired property of a adversarial loss)**.** *For any data distribution $P_d$, the generator loss $L_G$ has a unique global minimum at $P_g = P_d$ with respect to the model distribution $P_g$.*

We can see that Property 2 makes $L_G - L_G^*$ a divergence between $P_d$ and $P_g$ for any fixed $P_d$, where $L_G^* = L_G \big|_{P_g = P_d}$ is a constant term irrelevant to optimization. In contrast, Property 1 provides a weaker version when the uniqueness of the global minimum is not guaranteed.

## 4.3 The $\Psi$ and $\psi$ functions

In order to derive the necessary and sufficient conditions for Property 1 and Property 2, we first observe from Equation (15) that

$$L_G = \max_D \; \int_{\boldsymbol{x}} P_d(\boldsymbol{x}) \, f(D(\boldsymbol{x})) + P_g(\boldsymbol{x}) \, g(D(\boldsymbol{x})) \, d\boldsymbol{x} \,, \tag{16}$$

Assuming $D$ can be any function that maps from $\mathcal{X}$ to $\mathbb{R}$,[4] we can let $y = D(\boldsymbol{x})$ be a variable independent of $\boldsymbol{x}$. Then, we can exchange the order of the integral and maximum in Equation (16) and obtain

$$L_G = \int_{\boldsymbol{x}} \max_D \; \Big( P_d(\boldsymbol{x}) \, f(D(\boldsymbol{x})) + P_g(\boldsymbol{x}) \, g(D(\boldsymbol{x})) \Big) \, d\boldsymbol{x} \,, \tag{17}$$

---

[4]This assumption does not hold if $D$ is parameterized as a neural network. See the discussions in Section 4.7.

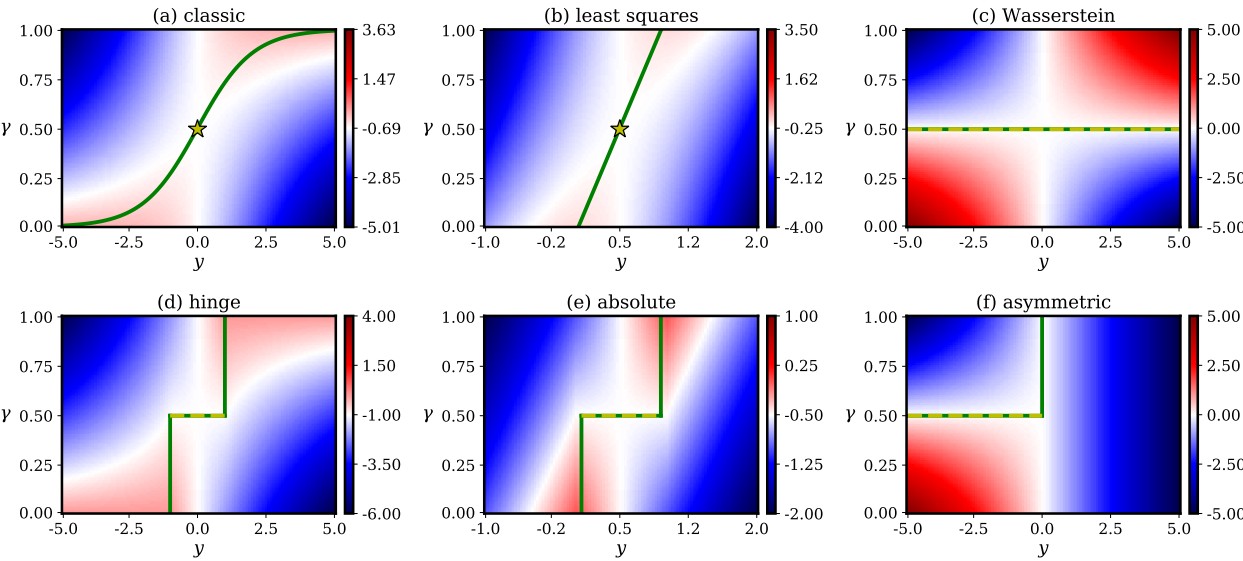

Figure 2: Graphs of the corresponding $\Psi$ functions for the adversarial losses discussed in this paper (see Equation (20) for the definition of $\Psi$). The green lines show the domains of $\psi$, i.e., the values that $y$ can take for different $\gamma$. The stars and the yellow dashed lines indicate the global minima of $\psi$. The midpoint of each color map is intentionally set to the minimum of $\psi$, i.e., the value taken at the star marks or the yellow segments. Note that as $y \in \mathbb{R}$, we plot different portions of $y$ where the characteristics of $\Psi$ can be best observed. (Best viewed in color.)

Next, rearranging Equation (17) we obtain

$$L_G = \int_{\boldsymbol{x}} \left( P_d(\boldsymbol{x}) + P_g(\boldsymbol{x}) \right) \max_D \left( \frac{P_d(\boldsymbol{x})\, f(D(\boldsymbol{x}))}{P_d(\boldsymbol{x}) + P_g(\boldsymbol{x})} + \frac{P_g(\boldsymbol{x})\, g(D(\boldsymbol{x}))}{P_d(\boldsymbol{x}) + P_g(\boldsymbol{x})} \right) \, d\boldsymbol{x} \,. \tag{18}$$

Now, let $y = D(\boldsymbol{x})$ and $\gamma = \frac{P_d(\boldsymbol{x})}{P_d(\boldsymbol{x}) + P_g(\boldsymbol{x})}$ (note that $\gamma = \frac{1}{2}$ if and only if $P_d(\boldsymbol{x}) = P_g(\boldsymbol{x})$). We then obatin

$$L_G = \int_{\boldsymbol{x}} \left( P_d(\boldsymbol{x}) + P_g(\boldsymbol{x}) \right) \max_y \; \gamma\, f(y) + (1 - \gamma)\, g(y) \, d\boldsymbol{x} \,. \tag{19}$$

Let us consider the latter term inside the integral and define the following two functions:

$$\Psi(\gamma, y) = \gamma\, f(y) + (1 - \gamma)\, g(y) \,, \tag{20}$$
$$\psi(\gamma) = \max_y \; \Psi(\gamma, y) \,, \tag{21}$$

where $\gamma \in [0, 1]$ and $y \in \mathbb{R}$. We visualize in Figure 2 the $\Psi$ functions and Figure 3 the $\psi$ functions for the adversarial losses considered in this paper.

## 4.4 Necessary and sufficient conditions for a well-behaved adversarial loss

With the $\psi$ function defined by Equation (21), we can now derive the necessary and sufficient conditions for a well-behaved adversarial loss. All proofs can be found in Appendix A. First, Theorems 1 and 2 provide the necessary conditions for a well-behaved adversarial loss.

**Theorem 1** (Necessary conditions of the weak desired property of an adversarial loss)**.** *If Property 1 holds, then for any $\gamma \in [0, 1]$, $\psi(\gamma) + \psi(1 - \gamma) \geq 2\,\psi(\frac{1}{2})$.*

**Theorem 2** (Necessary conditions of the strong desired property of an adversarial loss)**.** *If Property 2 holds, then for any $\gamma \in [0, \frac{1}{2}) \cup (\frac{1}{2}, 1]$, $\psi(\gamma) + \psi(1 - \gamma) > 2\,\psi(\frac{1}{2})$.*

For the sufficient conditions, we have Theorems 3 and 4 as follows.

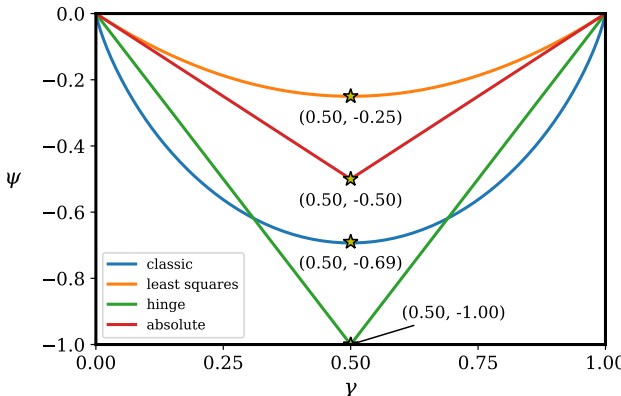

Figure 3: Graphs of the corresponding $\psi$ functions for the adversarial losses discussed in this paper (see Equation (21) for the definition of $\psi$). The stars indicate their global minima. For the Wasserstein loss, $\psi$ is only defined at $\gamma = 0.5$ as $\Psi(\gamma, y)$ is unbounded when $\gamma \neq 0.5$, where it takes a value of zero, and thus we do not include it in this plot.

**Theorem 3** (Sufficient conditions of the weak desired property of an adversarial loss). *If $\psi(\gamma)$ has a global minimum at $\gamma = \frac{1}{2}$, then Property 1 holds.*

**Theorem 4** (Sufficient conditions of the strong desired property of an adversarial loss). *If $\psi(\gamma)$ has a unique global minimum at $\gamma = \frac{1}{2}$, then Property 2 holds.*

With Theorems 3 and 4, we can check if a pair of output activation functions $f$ and $g$ form a well-behaved adversarial loss by finding their corresponding $\psi$ function.[5] Further, we have Theorem 5 for a more specific guideline for choosing or designing the output activation functions $f$ and $g$.

**Theorem 5** (Alternative sufficient conditions of the strong desired property of an adversarial loss). *If (1) $f'' + g'' \leq 0$ and (2) there exists some $y^* \in \mathbb{R}$ such that $f(y^*) = g(y^*)$ and $f'(y^*) = -g'(y^*) \neq 0$, then Property 2 holds.*

In other word, a pair of output activation functions $f$ and $g$ form a well-behaved adversarial loss as long as (1) the sum of $f$ and $g$ is concave downward and (2) we can find a point where they take the same value while having exact opposite nonzero derivatives. Intuitively, the second part of the prerequisites reflects the adversarial nature of the game between the discriminator and generator as it requires their objectives are opposite at some $y^* \in \mathbb{R}$.

Theorem 5 reveals a large class of theoretically valid adversarial losses. While such a theoretical analysis has not been done on the generalized form of adversarial losses in Equations (12) and (13), it happens that all the adversarial losses listed in Table 2 have such desirable properties, and we show in Table 2 their corresponding $y^*$ in Theorem 5. Moreover, we note that Nowozin et al. (2016) also derived a large class of adversarial losses based on the $f$-divergence. However, in their formulations, $f$ and $g$ need to be the conjugate function of each other. We discuss the connections to $f$-GANs in Appendix B Similarly, IPM-based GAN (Arjovsky et al., 2017) require $f$ and $g$ to be the same. In contrast, the prerequisites of Theorem 5 is less restricted. For example, the hinge loss (Lim & Ye, 2017; Tran et al., 2017) does not fall into either the $f$-GAN or IPM-based GAN classes, while it satisfies the prerequisites of Theorem 5 and is thus a well-behaved adversarial loss based on our theoretical results.

### 4.5 Analyzing the adversarial game by the $\Psi$ functions

Apart from deriving the necessary and sufficient conditions for a well-behaved adversarial loss, the $\Psi$ and $\psi$ functions defined in Equations (20) and (21) also provide some new insights into the adversarial behaviors of the generator and discriminator. If we follow Equation (19) and consider $y = D(\boldsymbol{x})$ and $\gamma(\boldsymbol{x}) = \frac{P_d(\boldsymbol{x})}{P_d(\boldsymbol{x}) + P_g(\boldsymbol{x})}$,

---

[5]Note that Theorem 4 still holds for the Wasserstein loss even if its $\psi$ function is only defined at $\gamma = 0.5$.

then the discriminator can be viewed as maximizing $\Psi$ along the $y$-axis in Figure 2. On the other hand, since the generator is trained to push $P_g$ towards $P_d$, it can be viewed as minimizing $\Psi$ along the $\gamma$-axis. Following this view, the saddle-shaped landscape of the $\Psi$ functions presented in Figure 2 reflects the minimax nature of the adversarial game. Moreover, they all have saddle points at $\gamma = \frac{1}{2}$, which happens when $P_d(\boldsymbol{x}) = P_g(\boldsymbol{x})$. In the ideal case when the discriminator can be trained till optimality in each iteration, we will always stay on the domain of $\psi$, i.e., the green line. In this case, the generator can be viewed as minimizing $\Psi$ along the green line.[6] Finally, the neighborhoods around the saddle points of $\Psi$ for different adversarial losses possess distinct characteristics, and such difference may affect the game dynamics and optimization stability. Further investigation on the properties of $\Psi$ and $\psi$ may lead insights into why and how different output activation functions perform differently in practice, which we will leave as future work.

### 4.6 Designing adversarial losses by the $\Psi$-landscape

From the analysis presented in the previous sections, we see that the $\Psi$ function reflects some important characteristics of an adversarial loss. By observing and designing the landscape of the $\Psi$ function, we propose the following two new adversarial losses that comply with Theorem 5.

- First, the *absolute loss* is obtained by replacing the square functions in the least squares loss with absolute functions. Specifically, we have $f(y) = -h(y) = -|1-y|$, $g(y) = -|y|$. As shown in Figure 2(e), its $\Psi$-landscape features a bounded positive-valued region (i.e., the red-colored region) of similar shape to that of the least squares loss. It also has a bounded global minima region (i.e., the yellow dashed lines) similar to that of the hinge loss. We propose this loss as its output activation functions are simple, piecewise linear.

- Second, the *asymmetric loss* is obtained by negating the output activation function $f$ of a Wasserstein for $y \in \mathbb{R}^+$. Specifically, we have $f(y) = -|y|$, $g(y) = h(y) = -y$. As can be seen from Figure 2(f), it features an asymmetric, non-saddle shaped $\Psi$-landscape distinct from other adversarial losses. Its $\Psi$-landscape is similar to that of the Wasserstein loss, but the positive part of $y$ is all negative-valued. As all adversarial losses in Table 2 has symmetric $f$ and $g$, we propose this loss function to show that the output activation functions $f$ and $g$ for a well-behaved adversarial loss does not necessarily need to be symmetric.

We will include these two new losses in our empirical study in Section 6.2. While these two losses do not originate from any underlying divergence, we will show in Section 6.2 that they can achieve similar empirical performance against other adversarial losses, and that an adversarial loss does not necessarily need to have a saddle-shaped $\Psi$-landscape.

### 4.7 Limitations of the theoretical results

We want to discuss some of the limitations of our theoretical analysis presented in the previous sections. First, while our analysis originates from a variational divergence minimization perspective, the GAN training dynamic in practice has been shown to be closer to approaching a Nash equilibrium of the adversarial game than minimizing a specific divergence, as suggested by Fedus et al. (2018). While Theorem 5 reveals a large class of theoretically valid adversarial losses, their optimization and convergence properties remain unclear and require further investigations. Second, our derivation in Section 4.3 is based on the assumption that the discriminator $D$ can be any function. However, this does not hold in general. In practice, the discriminator is usually parameterized as a neural network, thus having limited expressiveness. Moreover, the regularization approaches adopted may further constraint the set of functions that $D$ can possibly be. For example, IPM-based GANs consider only 1-Lipschitz functions for $D$ (Arjovsky et al., 2017). Finally, we note that the sufficient conditions provided in Theorem 5 are not tight. To examine the tightness of these sufficient conditions, we will empirically examine in Section 6.6 some cases when the prerequisites of Theorem 5 do not hold.

---

[6]Since $L_G$ is an integral over all possible $\boldsymbol{x} \in \mathcal{X}$, this adversarial game is played in a high dimensional space.

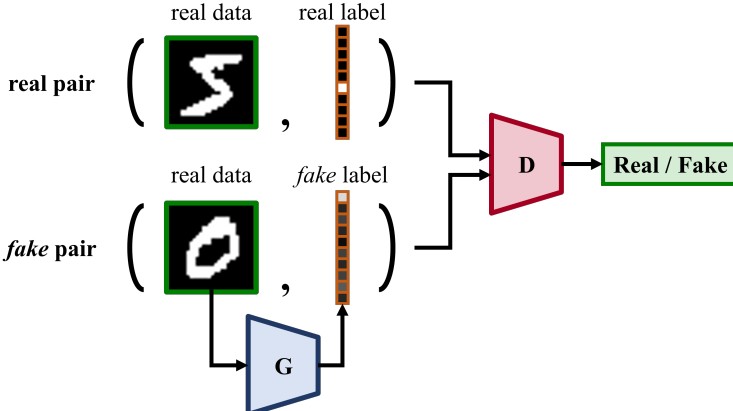

Figure 4: An example of a discriminative adversarial network (DAN) for MNIST digit classification.

## 5 A Comparative Framework for Adversarial Losses

In Section 4, we derive the necessary and sufficient conditions for a well-behaved adversarial loss. While the analysis reveals a large class of theoretical valid adversarial losses, it brings us to the next question—how does these different adversarial losses perform against one another? Although we focus only on the output activation functions in our theoretical analysis, a pair of adversarial losses can differ not only in their output activation functions but also the regularization approaches adopted. While Fedus et al. (2018); Lucic et al. (2018); Kurach et al. (2019) has attempted to decouple their effects, they evaluate only five, seven and ten, respectively, combinations of output activation functions and regularization approaches in their experiments due to the high computational cost for conducting the comparative study. This necessitates the needs for an efficient way to empirically compare different adversarial losses. Motivated by the various domains that the adversarial losses have been applied to, we propose to compare adversarial losses through using discriminative adversarial networks (DANs).

In this paper, we will work on the MNIST handwritten digit classification problem (LeCun et al., 1998). As illustrated in Figure 4, the generator $G$ in a DAN takes a sample $x \in \mathcal{X}$ as input and predict a fake label $c \in C$, where $C$ is the set of all possible classes. The discriminator takes a pair $(x, c)$ ($x \in \mathcal{X}$, $c \in C$) as input, where the label can either be the real label or a label predicted by the generator, and aims to examine if it is a real pair. Mathematically, the discriminator of a DAN learns the same mapping $\mathcal{X} \times \mathcal{Y} \mapsto \mathbb{R}$ as that of a conditional GAN does. The generator, a classification model in this example, is then trained by the supervision signal provided by the discriminator.

Although we work on a classification task in this paper, the proposed framework is generic to the underlying task as long as the evaluation metrics for the task are well defined. Moreover, adopting a DAN in this framework offers two additional advantages over a GAN. First, it avoids the high computation cost for training a GAN as the generator in a DAN is a discriminative model rather than a generative model in a GAN. Second, we can use standard evaluation metrics to evaluate a DAN. Hence, its performance can be easier evaluated than a GAN, which is usually assessed via more complex metrics such as the Inception Score (Salimans et al., 2016) and Fréchet Inception Distance (Heusel et al., 2017).

## 6 Empirical Results

Based on the comparative framework proposed in Section 5, we present in this section our empirical results on Question 2: How different combinations of output activation functions and regularization approaches perform empirically against one another? In addition, we use the proposed framework to examine the effects of penalty weights, Lipschitz constants and momentum terms in the optimizer. Moreover, we examine the tightness of the sufficient conditions provided by Theorem 5 and the robustness of different regularization approaches to data imbalance.

Table 3: Network architectures for the (a) generator $G$ and (b) discriminator $D$ used in our experiments.

| (a) Generator | | | | (b) Discriminator | | | |
|---|---|---|---|---|---|---|---|
| Layer type | Filters/neurons | Kernel | Stride | Layer type | Filters/neurons | Kernel | Stride |
| Convolutional | 32 | $3 \times 3$ | $3 \times 3$ | Convolutional | 32 | $3 \times 3$ | $3 \times 3$ |
| Convolutional | 64 | $3 \times 3$ | $3 \times 3$ | Convolutional | 64 | $3 \times 3$ | $3 \times 3$ |
| Max pooling | - | $2 \times 2$ | $2 \times 2$ | Max pooling | - | $2 \times 2$ | $2 \times 2$ |
| Dense | 128 | - | - | Dense | 128 | - | - |
| Dense | 10 | - | - | Dense | 1 | - | - |

## 6.1 Implementation details

We implement the comparative framework described in Section 5 on the MNIST handwritten digit classification problem (LeCun et al., 1998). We implement both the generator $G$ and discriminator $D$ as convolutional neural networks (CNNs). In addition, we use the same network architectures across all experiments, as summarized in Table 3. Moreover, we concatenate the label vector to the input of each layer in the discriminator. To speed up and stabilize the training, we use the batch normalization (Ioffe & Szegedy, 2015) in the generator, and the layer normalization (Ba et al., 2016) in the discriminator, which is disabled when the spectral normalization (Miyato et al., 2018) is used. For all the gradient penalties, we use Euclidean norms and a penalty weight $\lambda = 10$ in Equation (11). In addition, we use a Lipschitz constant $k = 1$ for the coupled and local gradient penalties (Gulrajani et al., 2017; Kodali et al., 2017). We set the hyperparameter $c$ for the local gradient penalties (Kodali et al., 2017) to 0.01 (see Table 1). We use ReLUs (Glorot et al., 2011) as the activation functions for all layers except the last layer of the generator, which uses a softmax function, and the last layer of the discriminator, which has no activation function.[7]. For the training, we use the Adam optimizer (Kingma & Ba, 2014) with $\alpha = 0.001$, $\beta_1 = 0$ and $\beta_2 = 0.9$. We implement all the models in Python and TensorFlow (Abadi et al., 2016). All the experiments are run on NVIDIA 1080 Tis with a batch size of 64. We alternatively update $G$ and $D$ once in each iteration and train the model for 100,000 generator steps, which takes roughly two hours on an NVIDIA 2080 Ti. Thanks to the speed, we repeat each experiment for ten runs to reduce the effects of randomness on our experimental results. For evaluation, we report the mean error rate and its 95% confidence interval. For reproducibility, we will release all the source code upon acceptance.

## 6.2 Decoupling the effects of output activation functions and regularization approaches

In the first experiment, we aim to study how different output activation functions and regularization approaches perform against one another. Moreover, in order to decouple their effects, we evaluate a comprehensive set of 168 combinations of twelve output activation functions and fourteen regularization approaches. For the output activation functions, we consider the classic minimax and nonsaturating losses (Goodfellow et al., 2014), Wasserstein loss (Arjovsky et al., 2017),[8] least squares loss (Mao et al., 2017), hinge loss (Lim & Ye, 2017; Tran et al., 2017), relativistic average and relativistic average hinge losses (Jolicoeur-Martineau, 2019), as well as the absolute and asymmetric losses we propose in Section 4.5. Moreover, while our theoretical analysis focuses only on the output activation functions $f$ and $g$, we also aim to examine the effects of the output activation function $h$ for the generator. Specifically, we examine pairing the classic and hinge losses with the following three variants of output activation function $h$: *minimax*—$h(y) = g(y)$, *nonsaturating*—$h(y) = \log(1 + e^{-y})$, and *linear*—$h(y) = -y$. For the regularization approaches, we consider the coupled gradient penalties (Gulrajani et al., 2017), local gradient penalties (Kodali et al., 2017), $R_1$ and $R_2$ gradient penalties (Mescheder et al., 2018) and the spectral normalization (Miyato et al., 2018). For the coupled and

---

[7]We do not apply any activation function to the output layer of the discriminator because the activation function is instead implemented in the loss terms in our formulation (see Equations (12) and (13)).

[8]In this paper, we will refer to the Wasserstein loss as a type of output activation functions defined in Table 2. The Lipschitz constraints introduced by Arjovsky et al. (2017) are decoupled and imposed by different regularization approaches.

local gradient penalties, we examine both the two-side and one-side versions (see Table 1). In addition, we also examine the combinations of the spectral normalization with different gradient penalties.

We report in Table 4 the results of the comparative study.[9] Our empirical results suggest that *there is no single winning combination of output activation functions and regularization approach across all settings.* The global lowest error rate is achieved by the combination of the relativistic hinge loss and two-side local gradient penalty. However, either relativisitic hinge loss or two-side local gradient penalty does not always achieve the lowest error rate when combined with other output activation functions or regularization approaches. Moreover, some combinations result in a notable high error rates, e.g., "classic minimax loss + $R_1$ gradient penalty," "least squares loss + $R_1$ gradient penalty," and "least squares loss + $R_2$ gradient penalty."

*With respect to the output activation functions,* we find that the classic minimax and nonsaturating losses never achieve the lowest error rates with any regularization approach. In general, The hinge minimax and relativistic average hinge losses achieve lower error rates than other losses, achieving the lowest error rates for three and four regularization approaches, respectively. The relativistic average loss outperforms both the classic minimax and nonsaturating losses across all regularization approaches, while the relativistic average hinge loss does not always outperform the standard hinge losses. The proposed absolute and asymmetric losses are more robust to different regularization approaches, representing the only two losses that achieve a mean error rate under 12% with any regularization approach. Regarding the output activation function $h$ for the generator, the three variants perform similarly and the differences do not reach statistical significance for most regularization approaches except the unregularized case. For the unregaularized case, the minimax version significantly outperforms the other two variants.

*With respect to the regularization approaches,* we find that the coupled and local gradient penalties outperform the $R_1$ and $R_2$ gradient penalties across nearly all output activation functions, no matter whether the spectral normalization is used or not. The two-side local penalty achieves the lowest error rate for six output activation functions. Combining either the coupled or local gradient penalty with the spectral normalization usually leads to higher error rates than using the gradient penalty only. The one-side gradient penalties lead to higher error rates than their two-side counterparts for all output activation functions.

To better analyze the empirical results, we also report in Figure 5(a) the minimum mean error rates achieved for each output activation function across different regularization approaches. Similarly, Figure 5(b) the minimum mean error rates achieved for regularization approach across different output activation functions. First, the hinge loss significantly outperforms the classic loss, no matter what the generator loss functions are. Second, the least squares loss performs significantly worse than the other output activation functions, while the hinge, Wasserstein, relativistic and absolute losses perform similarly. Third, the two two-sided gradient penalties significantly outperform other gradient penalties, no matter whether the spectral normalization is applied or not. Finally, combining the spectral normalization with gradient penalties does not necessarily lead to lower error rates.

Similar to Figure 5, we also report in Figure 6 the maximum mean error rates achieved. Note that we exclude the unregularized versions in Figure 6 to provide more meaningful analysis as the unregularized versions tends to fail easily. This serves as an indicator to the robustness and stability of each output activation function and regularization approach when combined with different components. First, the least squares loss are not robust to regularization approaches.[10] Second, while the relativistic hinge loss achieves the global lowest error rates when combined with the two-way local gradient penalties, it is not as robust to the asymmetric loss, which achieves the second lowest error rates when combined with the two-way coupled gradient penalties. Third, the spectral normalization improves the robustness for all gradient penalties.

Further, to investigate how the training dynamic is affected by the regularization approach, we present in Figure 7 the training progress for the nonsaturating loss with different regularization approaches. We can see that the coupled and local gradient penalties result in a stable training dynamic, no matter whether the spectral normalization is used or not. The $R_1$ and $R_2$ gradient penalties are too strong and halt the training early. This is possibly because they encourage the discriminator to have small gradients, and thus the gradients for both the generator and discriminator may vanish when the data and model distributions are

---

[9]As a baseline, we train the generator $G$ with the cross entropy loss, and the minimum error rate it achieves is 4.12%.

[10]Similarly, Lucic et al. (2018) also found that the least-squared GAN is not robust to random weight initialization.

Table 4: Error rates (%) for different combinations of output activation functions and regularization approaches. Mean values over ten runs and 95% confidence intervals are reported. Underlined and bold fonts indicate the lowest error rates per column and per row, respectively. (Abbreviations: GP—gradient penalty, SN—spectral normalization, TCGP—two-side coupled gradient penalty, TLGP—two-side local gradient penalty, OCGP—one-side coupled gradient penalty, OLGP—one-side local gradient penalty.)

| | Unregularized | TCGP | TLGP | $R_1$ GP | $R_2$ GP |
|---|---|---|---|---|---|
| Classic minimax (2014) | $9.11 \pm 0.39$ | $5.65 \pm 0.17$ | $\mathbf{5.42 \pm 0.11}$ | $19.01 \pm 2.31$ | $12.91 \pm 0.70$ |
| Classic nonsaturating (2014) | $26.83 \pm 4.44$ | $5.64 \pm 0.14$ | $5.56 \pm 0.19$ | $14.67 \pm 3.01$ | $13.80 \pm 1.98$ |
| Classic linear | $17.38 \pm 3.20$ | $5.66 \pm 0.22$ | $5.55 \pm 0.10$ | $18.49 \pm 3.42$ | $14.92 \pm 3.22$ |
| Hinge minimax | $\underline{5.57 \pm 0.16}$ | $\mathbf{4.83 \pm 0.21}$ | $4.88 \pm 0.15$ | $\underline{7.31 \pm 0.92}$ | $9.49 \pm 3.28$ |
| Hinge nonsaturating | $37.55 \pm 12.53$ | $5.00 \pm 0.15$ | $\mathbf{4.97 \pm 0.15}$ | $7.34 \pm 1.13$ | $7.54 \pm 0.81$ |
| Hinge linear (2017; 2017) | $11.50 \pm 3.30$ | $5.01 \pm 0.16$ | $\mathbf{4.89 \pm 0.11}$ | $8.96 \pm 2.20$ | $7.71 \pm 1.13$ |
| Wasserstein (2017) | $7.69 \pm 0.20$ | $5.04 \pm 0.12$ | $\mathbf{4.92 \pm 0.14}$ | $13.89 \pm 12.79$ | $7.25 \pm 0.74$ |
| Least squares (2017) | $7.15 \pm 0.29$ | $7.27 \pm 0.27$ | $6.70 \pm 0.27$ | $30.12 \pm 17.62$ | $32.44 \pm 13.05$ |
| Relativistic average (2019) | $90.20 \pm 0.00$ | $5.25 \pm 0.15$ | $\mathbf{5.01 \pm 0.19}$ | $8.00 \pm 1.01$ | $8.75 \pm 3.61$ |
| Relativistic average hinge (2019) | $52.01 \pm 5.81$ | $8.28 \pm 6.36$ | $\underline{\mathbf{4.71 \pm 0.07}}$ | $8.39 \pm 1.19$ | $7.67 \pm 1.13$ |
| Absolute | $6.69 \pm 0.15$ | $5.23 \pm 0.18$ | $5.20 \pm 0.16$ | $8.01 \pm 1.21$ | $\underline{6.64 \pm 0.32}$ |
| Asymmetric | $7.81 \pm 0.17$ | $\underline{\mathbf{4.77 \pm 0.21}}$ | $4.94 \pm 0.09$ | $8.79 \pm 1.97$ | $7.33 \pm 0.63$ |

| | SN | SN + TCGP | SN + TLGP | SN + $R_1$ GP | SN + $R_2$ GP |
|---|---|---|---|---|---|
| Classic minimax (2014) | $7.37 \pm 0.32$ | $5.55 \pm 0.23$ | $5.57 \pm 0.17$ | $11.16 \pm 1.65$ | $14.00 \pm 1.54$ |
| Classic nonsaturating (2014) | $8.25 \pm 0.22$ | $\mathbf{5.52 \pm 0.10}$ | $5.61 \pm 0.31$ | $12.98 \pm 1.68$ | $13.50 \pm 2.34$ |
| Classic linear | $7.98 \pm 0.22$ | $5.70 \pm 0.22$ | $\mathbf{5.48 \pm 0.18}$ | $15.45 \pm 4.05$ | $17.61 \pm 4.71$ |
| Hinge minimax | $6.22 \pm 0.14$ | $\underline{4.93 \pm 0.12}$ | $5.06 \pm 0.20$ | $10.62 \pm 1.30$ | $12.91 \pm 2.66$ |
| Hinge nonsaturating | $6.90 \pm 0.20$ | $5.05 \pm 0.14$ | $5.06 \pm 0.24$ | $11.91 \pm 2.49$ | $12.10 \pm 2.93$ |
| Hinge linear (2017; 2017) | $6.59 \pm 0.19$ | $4.97 \pm 0.12$ | $5.18 \pm 0.17$ | $13.63 \pm 2.56$ | $11.35 \pm 2.11$ |
| Wasserstein (2017) | $\underline{5.89 \pm 0.16}$ | $5.50 \pm 0.11$ | $5.76 \pm 0.43$ | $13.74 \pm 3.39$ | $13.82 \pm 3.06$ |
| Least squares (2017) | $7.88 \pm 0.28$ | $\mathbf{6.69 \pm 0.15}$ | $7.11 \pm 0.23$ | $9.91 \pm 0.96$ | $11.56 \pm 2.54$ |
| Relativistic average (2019) | $7.14 \pm 0.24$ | $5.35 \pm 0.18$ | $5.25 \pm 0.16$ | $9.31 \pm 1.25$ | $\underline{8.62 \pm 0.37}$ |
| Relativistic average hinge (2019) | $6.44 \pm 0.10$ | $5.02 \pm 0.19$ | $\underline{5.03 \pm 0.13}$ | $12.56 \pm 2.74$ | $12.40 \pm 2.82$ |
| Absolute | $6.79 \pm 0.28$ | $5.23 \pm 0.08$ | $\mathbf{5.18 \pm 0.22}$ | $10.42 \pm 1.90$ | $9.93 \pm 1.41$ |
| Asymmetric | $5.98 \pm 0.25$ | $5.60 \pm 0.18$ | $5.82 \pm 0.27$ | $\underline{8.46 \pm 0.27}$ | $8.80 \pm 0.73$ |

| | OCGP | OLGP | SN + OCGP | SN + OLGP |
|---|---|---|---|---|
| Classic minimax (2014) | $7.15 \pm 0.48$ | $6.95 \pm 0.32$ | $7.16 \pm 0.16$ | $6.86 \pm 0.18$ |
| Classic nonsaturating (2014) | $7.20 \pm 0.24$ | $6.98 \pm 0.14$ | $7.47 \pm 0.38$ | $7.15 \pm 0.22$ |
| Classic linear | $7.12 \pm 0.38$ | $7.00 \pm 0.62$ | $7.29 \pm 0.22$ | $7.18 \pm 0.33$ |
| Hinge minimax | $5.82 \pm 0.19$ | $7.33 \pm 0.84$ | $5.80 \pm 0.15$ | $5.83 \pm 0.12$ |
| Hinge nonsaturating | $\underline{5.69 \pm 0.19}$ | $7.88 \pm 0.82$ | $5.92 \pm 0.22$ | $5.74 \pm 0.17$ |
| Hinge linear (2017; 2017) | $\underline{5.77 \pm 0.18}$ | $6.22 \pm 0.64$ | $5.77 \pm 0.19$ | $5.82 \pm 0.12$ |
| Wasserstein (2017) | $7.60 \pm 1.87$ | $13.34 \pm 0.92$ | $6.35 \pm 0.27$ | $6.06 \pm 0.28$ |
| Least squares (2017) | $7.99 \pm 0.22$ | $8.06 \pm 0.30$ | $8.43 \pm 0.31$ | $8.31 \pm 0.32$ |
| Relativistic average (2019) | $8.03 \pm 2.06$ | $9.41 \pm 1.80$ | $6.18 \pm 0.18$ | $6.03 \pm 0.15$ |
| Relativistic average hinge (2019) | $10.70 \pm 1.56$ | $14.17 \pm 1.11$ | $\underline{5.42 \pm 0.20}$ | $\underline{5.42 \pm 0.20}$ |
| Absolute | $5.95 \pm 0.12$ | $\underline{5.88 \pm 0.25}$ | $6.22 \pm 0.15$ | $6.08 \pm 0.20$ |
| Asymmetric | $5.85 \pm 0.22$ | $7.57 \pm 0.61$ | $6.21 \pm 0.21$ | $5.92 \pm 0.23$ |

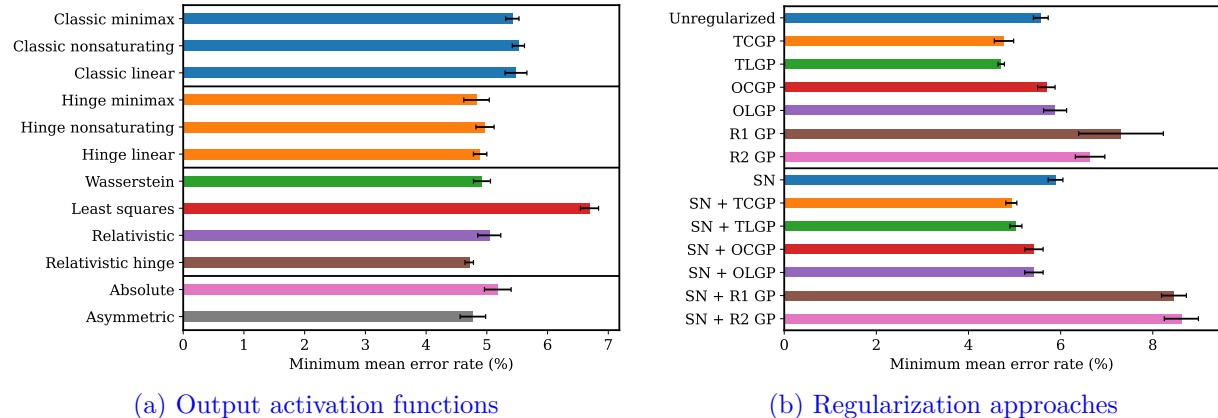

(a) Output activation functions    (b) Regularization approaches

Figure 5: Minimum mean error rates (over ten runs) achieved (a) for each output activation function across different regularization approaches and (b) for each regularization approach across different output activation functions. The error bars represented the 95% confidence intervals. (Abbreviations: GP—gradient penalty, SN—spectral normalization, TCGP—two-side coupled gradient penalty, TLGP—two-side local gradient penalty, OCGP—one-side coupled gradient penalty, OLGP—one-side local gradient penalty.)

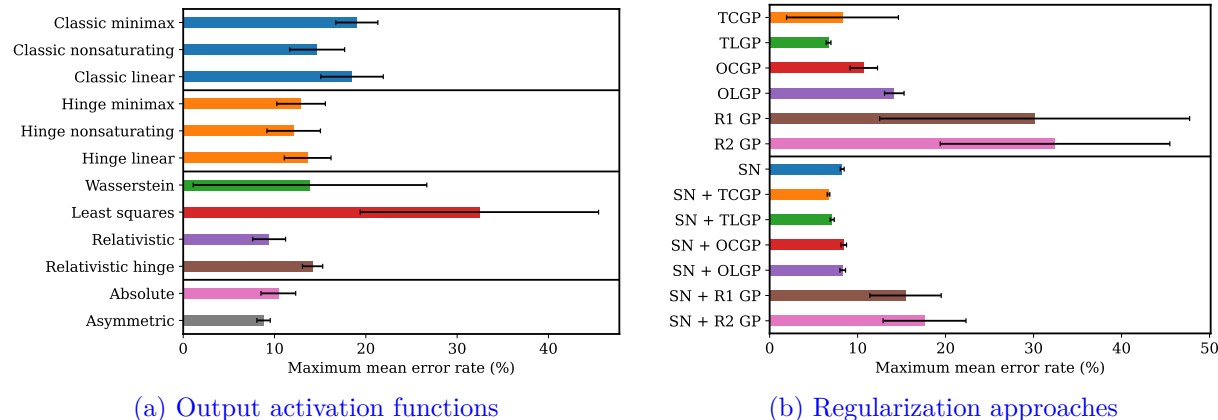

(a) Output activation functions    (b) Regularization approaches

Figure 6: Maximum mean error rates (over ten runs) achieved (a) for each output activation function across different regularization approaches and (b) for each regularization approach across different output activation functions. Both (a) and (b) exclude the unregularized versions. The error bars represented the 95% confidence intervals. (Abbreviations: GP—gradient penalty, SN—spectral normalization, TCGP—two-side coupled gradient penalty, TLGP—two-side local gradient penalty, OCGP—one-side coupled gradient penalty, OLGP—one-side local gradient penalty.)

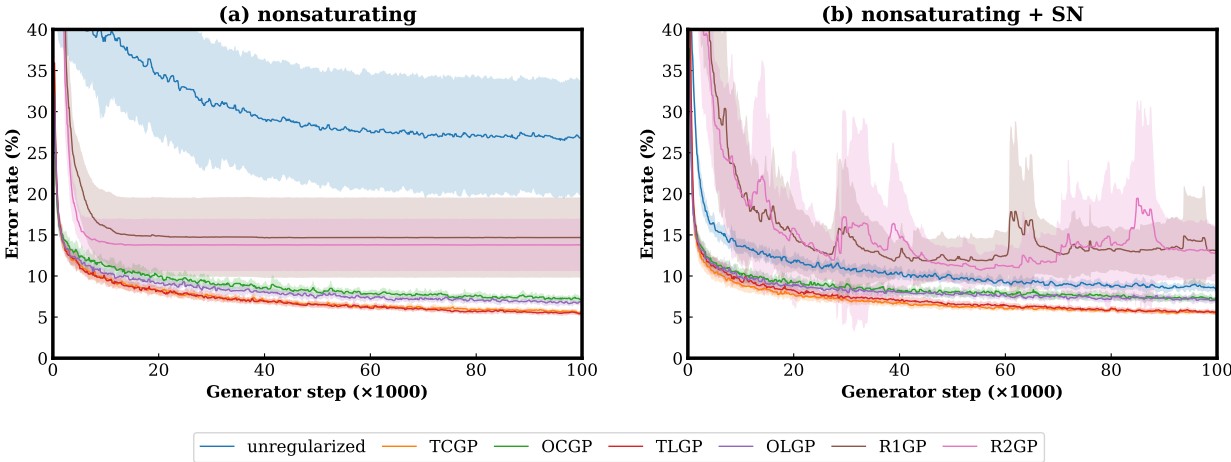

Figure 7: Error rates along the training progress for the classic nonsaturating loss with different regularization approaches. The models are evaluated every 100 steps, and the curves are smoothed by a median filter with a window size of 5. The shaded regions represent the standard deviations over ten runs. (Abbreviations: GP— gradient penalty, SN—spectral normalization, TCGP—two-side coupled gradient penalty, TLGP—two-side local gradient penalty.)

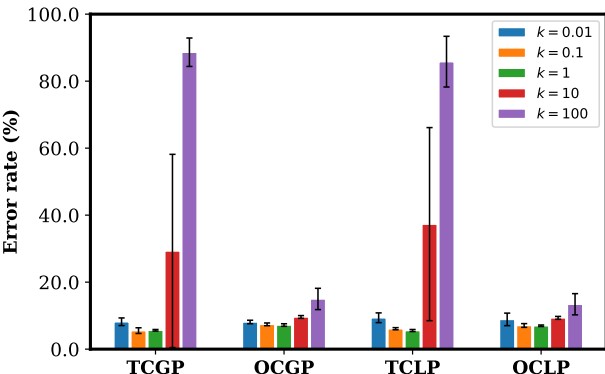

Figure 8: Error rates for different Lipschitz constants $k$ in the coupled and local gradient penalties, using the classic nonsaturating loss. Mean values over ten runs and standard deviations are reported. (Abbreviations: TCGP—two-side coupled gradient penalty, OCGP—one-side coupled gradient penalty, TLGP—two-side local gradient penalty, OLGP—one-side local gradient penalty.)

close enough. In addition, combining either the $R_1$ or $R_2$ gradient penalty with the spectral normalization leads to an unstable training dynamic.

## 6.3 Effects of the Lipschitz constants

In this experiment, we examine the effects of the Lipschitz constant $k$ in the coupled and local gradient penalties (Gulrajani et al., 2017; Kodali et al., 2017). We report in Figure 8 the results for $k = 0.01, 0.1, 1, 10, 100$ using the classic nonsaturating loss. We can see that the error rate increases as $k$ goes away from 1.0, suggesting that $k = 1$ is a good default value. In addition, the two-side gradient penalties achieve lower error rates when $k$ is set properly, while they are more sensitive to $k$ than their one-side counterparts.

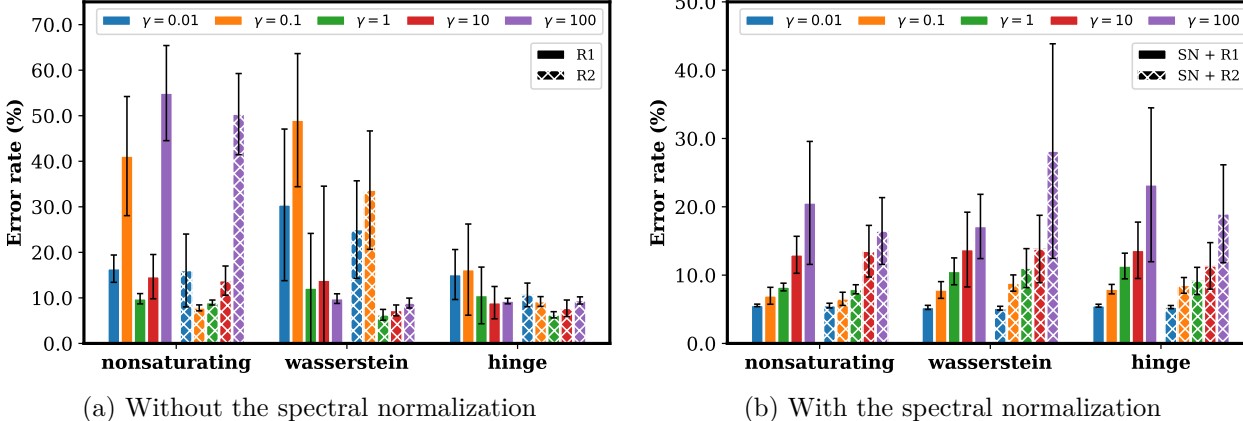

Figure 9: Error rates for different penalty weights $\lambda$ in the $R_1$ and $R_2$ gradient penalties. Mean values over ten runs and standard deviations are reported. (Abbreviation: SN—spectral normalization.)

## 6.4 Effects of the penalty weights

We examine in this experiment the effects of the penalty weights $\lambda$ for the $R_1$ and $R_2$ gradient penalties (Mescheder et al., 2018). We show in Figure 9 the results for $\lambda = 0.01, 0.1, 1, 10, 100$ using the classic nonsaturating, Wasserstein and hinge losses. We can see that the $R_2$ gradient penalty generally outperform the $R_1$ gradient penalty. However, they are both sensitive to the value of $\lambda$. Moreover, when the spectral normalization is used, the error rate increases as $\lambda$ increases, which implies that the $R_1$ and $R_2$ gradient penalties do not work well with the spectral normalization.

## 6.5 Effects of the momentum terms of the optimizers

We observe a trend in prior work towards using a smaller momentum (Radford et al., 2016), no momentum (Arjovsky et al., 2017; Gulrajani et al., 2017; Miyato et al., 2018; Brock et al., 2019) or a negative momentum (Gidel et al., 2019) in GAN training. Hence, we want to examine in this experiment the effects of the momentum term $\beta_1$ in an Adam optimizer (Kingma & Ba, 2014) with our proposed framework. Further, since the generator and discriminator are trained by distinct objectives, we also investigate setting different momentum values for them. We present in Figure 10 the results for all combinations of $\beta_1 = -0.5, 0.0, 0.5, 0.9$ for the generator and discriminator, using the classic nonsaturating loss along with the spectral normalization and coupled gradient penalties for regularization. We can see that for the two-side coupled gradient penalty, using a larger momentum in both the generator and discriminator leads to lower error rates. However, there is no specific trend for the one-side coupled gradient penalty.

## 6.6 Examining the necessary conditions for desirable adversarial loss functions

As discussed in Section 4.4, the sufficient conditions provided by Theorem 5 are not tight. That is, there are other valid output activation functions that satisfies Property 2 but not the conditions in Theorem 5. In order to examine the tightness of these sufficient conditions in Theorem 5, we examine in this experiment the cases when the prerequisites in Theorem 5 are violated. Specifically, we change the training objective for the discriminator into the following $\omega$-weighted loss:

$$\max_D \; \mathbb{E}_{\boldsymbol{x} \sim P_d}[\omega f(D(\boldsymbol{x}))] + \mathbb{E}_{\tilde{\boldsymbol{x}} \sim P_g}[g(D(\tilde{\boldsymbol{x}}))] , \tag{22}$$

where $\omega \in \mathbb{R}^+ \setminus \{1\}$ is a constant. Note that the training objective for the generator remains the same. We can see that the prerequisites in Theorem 5 do not hold when $\omega \neq 1$.[11] We consider the classic nonsaturating, the Wasserstein and hinge losses, using the spectral normalization for regularization. We show in Table 5

---

[11]See Appendix C for the graphs of the corresponding $\Psi$ and $\psi$ functions for $\omega = 0.5, 0.9, 1.0, 1.1, 2.0$.

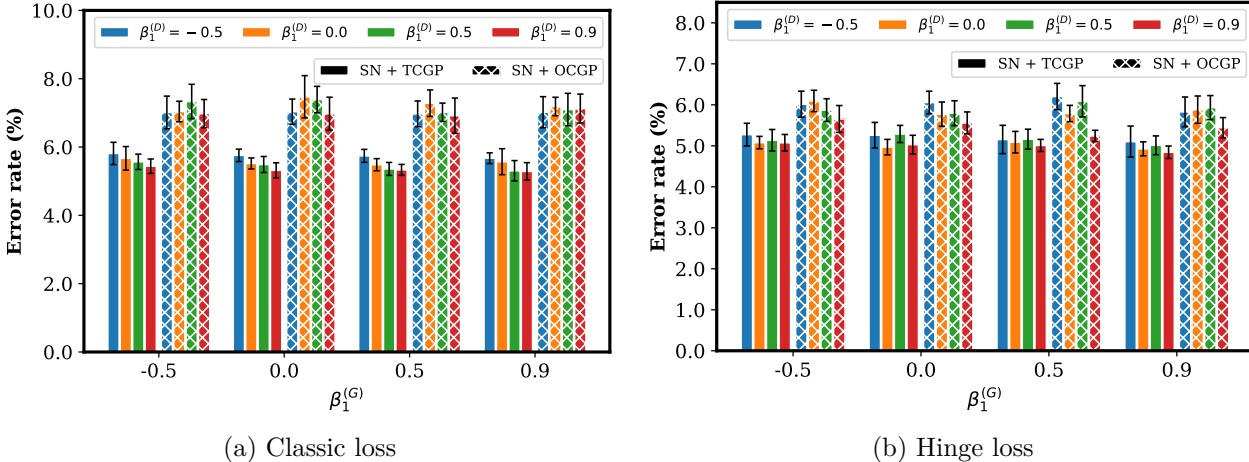

(a) Classic loss

(b) Hinge loss

Figure 10: Error rates for different momentum terms $\beta_1$ in the Adam optimizer. $\beta_1^{(G)}$ and $\beta_1^{(D)}$ denote the momentum terms for the generator and discriminator, respectively. Mean values over ten runs and standard deviations are reported. (Abbreviations: SN—spectral normalization, TCGP—two-side coupled gradient penalty, OCGP—one-side coupled gradient penalty; best viewed in color.)

Table 5: Error rates (%) for the nonsaturating, the Wasserstein and hinge losses along with their $\omega$-weighted versions (see Equation (22) for the definition). Mean values and 95% confidence intervals are reported.

|                | $\omega = 0.5$   | $\omega = 0.9$   | Original         | $\omega = 1.1$    | $\omega = 2.0$   |
|----------------|------------------|------------------|------------------|-------------------|------------------|
| Nonsaturating  | $8.47 \pm 0.22$  | $8.96 \pm 0.39$  | $\mathbf{8.25 \pm 0.22}$ | $8.62 \pm 0.28$  | $9.18 \pm 0.58$  |
| Wasserstein    | $73.16 \pm 3.94$ | $57.66 \pm 3.18$ | $\mathbf{5.89 \pm 0.16}$ | $60.30 \pm 4.72$ | $69.54 \pm 3.33$ |
| Hinge          | $15.20 \pm 1.52$ | $8.94 \pm 0.54$  | $\mathbf{6.59 \pm 0.19}$ | $8.02 \pm 0.22$  | $11.87 \pm 0.53$ |

the results for $\omega = 0.5, 0.9, 1.0, 1.1, 2.0$. We can see that the original losses result in the lowest error rates as compared to the $\omega$-weighted loss. While the Wasserstein loss fails with error rates over 50% when $\omega \neq 1$, the $\omega$-weighted versions of the nonsaturing and hinge losses achieve a higher yet still acceptable error rate. This suggests that even if the $\omega$-weighted loss violates the conditions in Theorem 5, it can still provide some supervision signal for training. Moreover, the error rates generally increase as $\omega$ goes away from 1.0. These results suggest that although Theorem 5 reveals a large class of theoretical valid adversarial losses, the sufficient conditions it provide are not tight.

### 6.7 Limitations of the empirical results

While the proposed comparative framework allows us to efficiently compare adversarial losses without high computation costs, one of the main limitations of our empirical results is that it is unclear whether the empirical results we obtain for DANs can be generalized to conditional and unconditional GANs. Although the discriminators in a DAN and a conditional GAN are trained to optimize the same objective, learning the same mapping $\mathcal{X} \times \mathcal{Y} \mapsto \mathbb{R}$, the generators are trained to learn the opposite mappings—the generator in a DAN learns the mapping $\mathcal{X} \mapsto \mathcal{Y}$, while the generator in a conditional GAN learns the mapping $\mathcal{Y} \mapsto \mathcal{X}$. Nonetheless, we argue that the idea of adversarial losses is rather generic and can be applied to various tasks, either generative or discriminative. Our empirical results still contribute to the goal towards a deeper understanding of adversarial losses.

Another limitation of our empirical study is the lack of hyperparameter search due to computation budget. In our experiments, we follow the recommended hyperparameter settings for each regularization approach. However, as suggested by Lucic et al. (2018) and Kurach et al. (2019), most GAN models can reach similar scores with enough hyperparameter optimization and random restarts. Accordingly, certain models might be

able to reach a lower error rate with proper hyperparameter optimizations. Nevertheless, we argue that this phenomenon is less presented in our case as we use the adversarial losses to train the generator to perform a much simpler task of handwritten digit classification. In general, we observe a small variance on most of our empirical results except certain combinations.

## 7 Conclusion

In the first part of this paper, following the perspective of variational divergence minimization, we derived the necessary and sufficient conditions of a well-behaved adversarial loss in a generalized formulation. Our theoretical analysis reveals a large class of adversarial losses that includes a range of adversarial losses proposed in prior work. In the second part of this paper, we proposed a simple comparative framework for adversarial losses based on discriminative adversarial networks. With the proposed framework, we evaluated a comprehensive set of 168 adversarial losses featuring the combinations of twelve output activation functions and fourteen regularization approaches. Our empirical findings suggest that there is no single winning combination of output activation functions and regularization approaches across all settings. The theoretical and empirical results presented in this paper may together serve as a reference for choosing or designing adversarial losses in future research.

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

# A  Proofs of Theorems

**Theorem 1** (Necessary conditions of the weak desired property of an adversarial loss)**.** *If Property 1 holds, then for any $\gamma \in [0, 1]$, $\psi(\gamma) + \psi(1 - \gamma) \geq 2\,\psi(\frac{1}{2})$.*

*Proof.* Since Property 1 holds, we have for any fixed $P_d$,

$$L_G \geq L_G \big|_{P_g = P_d}. \tag{23}$$

Let us consider

$$P_d(\boldsymbol{x}) = \gamma\,\delta(\boldsymbol{x} - \boldsymbol{s}) + (1 - \gamma)\,\delta(\boldsymbol{x} - \boldsymbol{t}), \tag{24}$$
$$P_g(\boldsymbol{x}) = (1 - \gamma)\,\delta(\boldsymbol{x} - \boldsymbol{s}) + \gamma\,\delta(\boldsymbol{x} - \boldsymbol{t}). \tag{25}$$

for some $\gamma \in [0, 1]$ and $\boldsymbol{s}, \boldsymbol{t} \in \mathcal{X}, \boldsymbol{s} \neq \boldsymbol{t}$. Then, we have

$$L_G \big|_{P_g = P_d} = \max_D \int_{\boldsymbol{x}} P_d(\boldsymbol{x})\,f(D(\boldsymbol{x})) + P_d(\boldsymbol{x})\,g(D(\boldsymbol{x}))\,d\boldsymbol{x} \tag{26}$$

$$= \max_{D} \int_{\boldsymbol{x}} P_d(\boldsymbol{x}) \left( f(D(\boldsymbol{x})) + g(D(\boldsymbol{x})) \right) d\boldsymbol{x} \tag{27}$$

$$= \max_{D} \int_{\boldsymbol{x}} \left( (\gamma \, \delta(\boldsymbol{x} - \boldsymbol{s}) + (1 - \gamma) \, \delta(\boldsymbol{x} - \boldsymbol{t}))(f(D(\boldsymbol{x})) + g(D(\boldsymbol{x}))) \right) d\boldsymbol{x} \tag{28}$$

$$= \max_{D} \left( \gamma \, (f(D(\boldsymbol{s})) + g(D(\boldsymbol{s}))) + (1 - \gamma)(f(D(\boldsymbol{t})) + g(D(\boldsymbol{t}))) \right) \tag{29}$$

$$= \max_{y_1, y_2} \gamma \, (f(y_1) + g(y_1)) + (1 - \gamma)(f(y_2) + g(y_2)) \tag{30}$$

$$= \max_{y_1} \gamma \, (f(y_1) + g(y_1)) + \max_{y_2} (1 - \gamma)(f(y_2) + g(y_2)) \tag{31}$$

$$= \max_{y} f(y) + g(y) \tag{32}$$

$$= 2 \, \psi(\tfrac{1}{2}) \,. \tag{33}$$

Moreover, we have

$$L_G = \max_{D} \int_{\boldsymbol{x}} P_d(\boldsymbol{x}) \, f(D(\boldsymbol{x})) + P_g(\boldsymbol{x}) \, g(D(\boldsymbol{x})) \, d\boldsymbol{x} \tag{34}$$

$$= \max_{D} \int_{\boldsymbol{x}} \left( (\gamma \, \delta(\boldsymbol{x} - \boldsymbol{s}) + (1 - \gamma) \, \delta(\boldsymbol{x} - \boldsymbol{t})) \, f(D(\boldsymbol{x})) + ((1 - \gamma) \, \delta(\boldsymbol{x} - \boldsymbol{s}) + \gamma \, \delta(\boldsymbol{x} - \boldsymbol{t})) \, g(D(\boldsymbol{x})) \right) d\boldsymbol{x} \tag{35}$$

$$= \max_{D} \left( \gamma \, f(D(\boldsymbol{s})) + (1 - \gamma) \, f(D(\boldsymbol{t})) + (1 - \gamma) \, g(D(\boldsymbol{s})) + \gamma \, g(D(\boldsymbol{t})) \right) \tag{36}$$

$$= \max_{y_1, y_2} \left( \gamma \, f(y_1) + (1 - \gamma) \, g(y_1)) + (1 - \gamma) \, f(y_2) + \gamma \, g(y_2) \right) \tag{37}$$

$$= \max_{y_1} \gamma \, f(y_1) + (1 - \gamma) \, g(y_1)) + \max_{y_2} (1 - \gamma) \, f(y_2) + \gamma \, g(y_2) \tag{38}$$

$$= \psi(\gamma) + \psi(1 - \gamma) \,. \tag{39}$$

(Note that we can obtain Equation (30) from Equation (29) and Equation (37) from Equation (36) because $D$ can be any function and thus $D(\boldsymbol{s})$ is independent of $D(\boldsymbol{t})$.)

As Equation (23) holds for any fixed $P_d$, by substituting Equation (33) and Equation (38) into Equation (23), we get

$$\psi(\gamma) + \psi(1 - \gamma) \geq 2 \, \psi(\tfrac{1}{2}) \tag{40}$$

for any $\gamma \in [0, 1]$, which concludes the proof. $\qquad\square$

**Theorem 2** (Necessary conditions of the strong desired property of an adversarial loss)**.** *If Property 2 holds, then for any* $\gamma \in [0, \frac{1}{2}) \cup (\frac{1}{2}, 1]$, $\psi(\gamma) + \psi(1 - \gamma) > 2 \, \psi(\frac{1}{2})$.

*Proof.* Since Property 2 holds, we have for any fixed $P_d$,

$$L_G \big|_{P_g \neq P_d} > L_G \big|_{P_g = P_d} \,. \tag{41}$$

Following the proof of Theorem 1, consider

$$P_d(\boldsymbol{x}) = \gamma \, \delta(\boldsymbol{x} - \boldsymbol{s}) + (1 - \gamma) \, \delta(\boldsymbol{x} - \boldsymbol{t}) \,, \tag{42}$$

$$P_g(\boldsymbol{x}) = (1 - \gamma) \, \delta(\boldsymbol{x} - \boldsymbol{s}) + \gamma \, \delta(\boldsymbol{x} - \boldsymbol{t}) \,, \tag{43}$$

for some $\gamma \in [0, 1]$ and some $\boldsymbol{s}, \boldsymbol{t} \in \mathcal{X}, \boldsymbol{s} \neq \boldsymbol{t}$. It can be easily shown that $P_g = P_d$ if and only if $\gamma = \frac{1}{2}$.

As Equation (41) holds for any fixed $P_d$, by substituting Equation (42) and Equation (43) into Equation (41), we get

$$\psi(\gamma) + \psi(1 - \gamma) > 2 \, \psi(\tfrac{1}{2}) \,, \tag{44}$$

for any $\gamma \in [0, \frac{1}{2}) \cup (\frac{1}{2}, 1]$, concluding the proof. $\qquad\square$

**Theorem 3** (Sufficient conditions of the weak desired property of an adversarial loss)**.** *If* $\psi(\gamma)$ *has a global minimum at* $\gamma = \frac{1}{2}$, *then Property 1 holds.*

*Proof.* First, we see that

$$L_G\big|_{P_g=P_d} = \max_D \int_{\boldsymbol{x}} P_d(\boldsymbol{x})\,f(D(\boldsymbol{x})) + P_d(\boldsymbol{x})\,g(D(\boldsymbol{x}))\,d\boldsymbol{x} \tag{45}$$

$$= \max_y \int_{\boldsymbol{x}} P_d(\boldsymbol{x})\,f(y) + P_d(\boldsymbol{x})\,g(y)\,d\boldsymbol{x} \tag{46}$$

$$= \max_y \int_{\boldsymbol{x}} P_d(\boldsymbol{x})\,\big(f(y) + g(y)\big)\,d\boldsymbol{x} \tag{47}$$

$$= \max_y \big(f(y) + g(y)\big) \int_{\boldsymbol{x}} P_d(\boldsymbol{x})\,d\boldsymbol{x} \tag{48}$$

$$= \max_y \; f(y) + g(y) \tag{49}$$

$$= 2\,\psi(\tfrac{1}{2})\,. \tag{50}$$

On the other had, by definition we have

$$L_G = \max_D \int_{\boldsymbol{x}} P_d(\boldsymbol{x})\,f(D(\boldsymbol{x})) + P_g(\boldsymbol{x})\,g(D(\boldsymbol{x}))\,d\boldsymbol{x}\,. \tag{51}$$

Since $D$ can be any function that maps from $\mathcal{X}$ to $\mathbb{R}$,[12] we can let $y = D(\boldsymbol{x})$ be a variable independent of $\boldsymbol{x}$. Thus, we have

$$L_G = \max_y \int_{\boldsymbol{x}} P_d(\boldsymbol{x})\,f(y) + P_g(\boldsymbol{x})\,g(y)\,d\boldsymbol{x} \tag{52}$$

$$= \max_y \int_{\boldsymbol{x}} \big(P_d(\boldsymbol{x}) + P_g(\boldsymbol{x})\big) \left( \frac{P_d(\boldsymbol{x})\,f(y)}{P_d(\boldsymbol{x}) + P_g(\boldsymbol{x})} + \frac{P_g(\boldsymbol{x})\,g(y)}{P_d(\boldsymbol{x}) + P_g(\boldsymbol{x})} \right) d\boldsymbol{x} \tag{53}$$

$$= \int_{\boldsymbol{x}} \big(P_d(\boldsymbol{x}) + P_g(\boldsymbol{x})\big) \max_y \left( \frac{P_d(\boldsymbol{x})\,f(y)}{P_d(\boldsymbol{x}) + P_g(\boldsymbol{x})} + \frac{P_g(\boldsymbol{x})\,g(y)}{P_d(\boldsymbol{x}) + P_g(\boldsymbol{x})} \right) d\boldsymbol{x}\,. \tag{54}$$

Since $\frac{P_d(\boldsymbol{x})}{P_d(\boldsymbol{x})+P_g(\boldsymbol{x})} \in [0,1]$, we have

$$L_G = \int_{\boldsymbol{x}} \big(P_d(\boldsymbol{x}) + P_g(\boldsymbol{x})\big)\,\psi\left( \frac{P_d(\boldsymbol{x})}{P_d(\boldsymbol{x}) + P_g(\boldsymbol{x})} \right) d\boldsymbol{x}\,. \tag{55}$$

As $\psi(\gamma)$ has a global minimum at $\gamma = \frac{1}{2}$, now we have

$$L_G \geq \int_{\boldsymbol{x}} \big(P_d(\boldsymbol{x}) + P_g(\boldsymbol{x})\big)\,\psi(\tfrac{1}{2})\,d\boldsymbol{x} \tag{56}$$

$$= \psi(\tfrac{1}{2}) \int_{\boldsymbol{x}} \big(P_d(\boldsymbol{x}) + P_g(\boldsymbol{x})\big)\,d\boldsymbol{x} \tag{57}$$

$$= 2\,\psi(\tfrac{1}{2})\,. \tag{58}$$

Finally, combining Equation (50) and Equation (58) yields

$$L_G \geq L_G\big|_{P_g=P_d}\,, \tag{59}$$

which holds for any $P_d$, thus concluding the proof. $\qquad\square$

**Theorem 4** (Sufficient conditions of the strong desired property of an adversarial loss). *If $\psi(\gamma)$ has a unique global minimum at $\gamma = \frac{1}{2}$, then Property 2 holds.*

*Proof.* Since $\psi(\gamma)$ has a unique global minimum at $\gamma = \frac{1}{2}$, we have for any $\gamma \in [0, \frac{1}{2}) \cup (\frac{1}{2}, 1]$,

$$\psi(\gamma) > \psi(\tfrac{1}{2})\,. \tag{60}$$

---

[12]Note that this is not always true if some constraints are imposed on $D$.

When $P_g \neq P_d$, there must be some $\boldsymbol{x}_0 \in \mathcal{X}$ such that $P_g(\boldsymbol{x}_0) \neq P_d(\boldsymbol{x}_0)$. Thus, $\frac{P_d(\boldsymbol{x}_0)}{P_d(\boldsymbol{x}_0)+P_g(\boldsymbol{x}_0)} \neq \frac{1}{2}$, and thereby $\psi\left(\frac{P_d(\boldsymbol{x}_0)}{P_d(\boldsymbol{x}_0)+P_g(\boldsymbol{x}_0)}\right) > \psi(\frac{1}{2})$. Now, by Equation (55) we have

$$L_G\big|_{P_g \neq P_d} = \int_{\boldsymbol{x}} \left(P_d(\boldsymbol{x}) + P_g(\boldsymbol{x})\right) \psi\left(\frac{P_d(\boldsymbol{x})}{P_d(\boldsymbol{x}) + P_g(\boldsymbol{x})}\right) d\boldsymbol{x} \tag{61}$$

$$> \int_{\boldsymbol{x}} \left(P_d(\boldsymbol{x}) + P_g(\boldsymbol{x})\right) \psi(\tfrac{1}{2}) \, d\boldsymbol{x} \tag{62}$$

$$= \psi(\tfrac{1}{2}) \int_{\boldsymbol{x}} \left(P_d(\boldsymbol{x}) + P_g(\boldsymbol{x})\right) d\boldsymbol{x} \tag{63}$$

$$= 2\, \psi(\tfrac{1}{2})\,. \tag{64}$$

Finally, combining Equation (50) and Equation (64) yields

$$L_G\big|_{P_g \neq P_d} > L_G\big|_{P_g = P_d}\,, \tag{65}$$

which holds for any $P_d$, thus concluding the proof. $\qquad\square$ $\qquad\qquad\square$

**Theorem 5** (Alternative sufficient conditions of the strong desired property of an adversarial loss). *If $f'' + g'' \leq 0$ and there exists some $y^*$ such that $f(y^*) = g(y^*)$ and $f'(y^*) = -g'(y^*) \neq 0$, then Property 2 holds.*

*Proof.* First, we have by definition

$$\Psi(\gamma, y) = \gamma\, f(y) + (1 - \gamma)\, g(y)\,. \tag{66}$$

By taking the partial derivatives, we get

$$\frac{\partial \Psi}{\partial \gamma} = f(y) - g(y)\,, \tag{67}$$

$$\frac{\partial \Psi}{\partial y} = \gamma\, f'(y) + (1 - \gamma)\, g'(y)\,, \tag{68}$$

$$\frac{\partial^2 \Psi}{\partial y^2} = \gamma\, f''(y) + (1 - \gamma)\, g''(y)\,. \tag{69}$$

We know that there exists some $y^*$ such that

$$f(y^*) = g(y^*)\,, \tag{70}$$

$$f'(y^*) = -g'(y^*) \neq 0\,. \tag{71}$$

(i) By Equation (67) and Equation (68), we see that

$$\frac{\partial \Psi}{\partial \gamma}\bigg|_{y=y^*} = 0\,, \tag{72}$$

$$\frac{\partial \Psi}{\partial y}\bigg|_{(\gamma,y)=(\frac{1}{2},y^*)} = 0\,. \tag{73}$$

Now, by Equation (72) we know that $\Psi$ is constant when $y = y^*$. That is, for any $\gamma \in [0, 1]$,

$$\Psi(\gamma, y^*) = \Psi(\tfrac{1}{2}, y^*)\,. \tag{74}$$

(ii) Because $f'' + g'' \leq 0$, by Equation (69) we have

$$\frac{\partial^2 \Psi}{\partial y^2}\bigg|_{\gamma=\frac{1}{2}} = \tfrac{1}{2} f''(y) + \tfrac{1}{2} g''(y) \tag{75}$$

$$\leq 0 \,. \tag{76}$$

By Equation (73) and Equation (75), we see that $y^*$ is a global minimum point of $\Psi\big|_{\gamma=\frac{1}{2}}$. Thus, we now have

$$\Psi(\tfrac{1}{2}, y^*) = \max_{y} \ \Psi(\tfrac{1}{2}, y) \tag{77}$$

$$= \psi(\tfrac{1}{2}) \,. \tag{78}$$

(iii) By Equation (68), we see that

$$\frac{\partial \Psi}{\partial y}\Big|_{y=y^*} = \gamma \, f'(y^*) + (1-\gamma) \, g'(y^*) \tag{79}$$

$$= \gamma \, f'(y^*) + (1-\gamma) \, (-f'(y^*)) \tag{80}$$

$$= (2\gamma - 1) \, f'(y^*) \,. \tag{81}$$

Since $f'(y^*) \neq 0$, we have

$$\frac{\partial \Psi}{\partial y}\Big|_{y=y^*} \neq 0 \quad \forall \, \gamma \in [0, \tfrac{1}{2}) \cup (\tfrac{1}{2}, 1] \,. \tag{82}$$

This shows that for any $\gamma \in [0, \tfrac{1}{2}) \cup (\tfrac{1}{2}, 1]$, there must exists some $y^\circ$ such that

$$\Psi(\gamma, y^\circ) > \Psi(\gamma, y^*) \,. \tag{83}$$

And by definition we have

$$\Psi(\gamma, y^\circ) < \max_{y} \ \Psi(\gamma, y) \tag{84}$$

$$= \psi(\gamma) \,. \tag{85}$$

Hence, by Equation (83) and Equation (84) we get

$$\psi(\gamma) > \Psi(\gamma, y^*) \,. \tag{86}$$

Now, combining Equation (73), Equation (78) and Equation (86) yields

$$\psi(\gamma) > \psi(\tfrac{1}{2}) \quad \forall \, \gamma \in [0, \tfrac{1}{2}) \cup (\tfrac{1}{2}, 1] \,. \tag{87}$$

Finally, by Theorem 4, we know that Property 2 holds, which concludes the proof. $\qquad\square$

## B  Connections to $f$-divergence

Nowozin et al. (2016) proposed the $f$-GAN family based on the $f$-divergence, which can be formulated as

$$L_G = \max_{D} \ \mathbb{E}_{\boldsymbol{x} \sim P_d}[D(\boldsymbol{x})] + \mathbb{E}_{\tilde{\boldsymbol{x}} \sim P_g}[-\hat{f}^*(D(\tilde{\boldsymbol{x}}))] \,, \tag{88}$$

where $\hat{f}$ is convex and lower-semicontinuous and $\hat{f}(1) = 0$.[13] Following the definition given by Equation (12), we have $f(y) = y$ and $g(y) = -\hat{f}^*(y)$. We show some representative $f$-GANs in Table 6 and their corresponding output activation functions.

To connect $f$-GANs to our theoretical results, we first derive their corresponding $\Psi$ and $\psi$ functions:

$$\Psi(\gamma, y) = \gamma \, f(y) + (1-\gamma) \, g(y) \tag{89}$$

$$= \gamma \, y - (1-\gamma) \, \hat{f}^*(y) \,. \tag{90}$$

---

[13]Here, we adopt the first formulation proposed by Nowozin et al. (2016), where the output activation $g_f$ is not included.

| Name | $\hat{f}$ | $g \ (= \hat{f}^*)$ | $g'$ | $y^*$ |
|---|---|---|---|---|
| Total variation | $|y - 1|/2$ | $-y$ | $-1$ | $0$ |
| Kullback–Leibler (KL) | $y \log y$ | $-e^{y-1}$ | $-e^{y-1}$ | - |
| Reverse KL | $-\log y$ | $1 + \log(-y)$ | $1/y$ | - |
| Pearson $\chi^2$ | $(1-y)^2$ | $-y - y^2/4$ | $-1 - y/2$ | $0$ |
| Neyman $\chi^2$ | $(1-y)^2/y$ | $-2 + 2\sqrt{1-y}$ | $-1/\sqrt{1-y}$ | $0$ |
| Squared Hellinger | $(1-\sqrt{y})^2$ | $-y/(1-y)$ | $-1/(1-y)^2$ | $0$ |
| Jensen-Shannon | $-(1+y)\log((1+y)/2) + y\log y$ | $\log(2 - e^y)$ | $-e^y/(2 - e^y)$ | $0$ |

Table 6: The corresponding output activation functions for different $f$-GANs.

| Name | $\psi(\gamma)$ | $\arg\min_\gamma \psi(\gamma)$ |
|---|---|---|
| Total variation | $|\gamma - 1/2|$ | $1/2$ |
| Kullback–Leibler (KL) | $\gamma \log(\gamma/(1-\gamma))$ | $\approx 0.2178$ |
| Reverse KL | $(1-\gamma)\log((1-\gamma)/\gamma)$ | $\approx 0.7822$ |
| Pearson $\chi^2$ | $(1-2\gamma)^2/(1-\gamma)$ | $1/2$ |
| Neyman $\chi^2$ | $(1-2\gamma)^2/\gamma$ | $1/2$ |
| Squared Hellinger | $1 - 2\sqrt{\gamma(1-\gamma)}$ | $1/2$ |
| Jensen-Shannon | $\log 2 + \gamma \log \gamma + (1-\gamma)\log(1-\gamma)$ | $1/2$ |

Table 7: The corresponding $\psi$ functions for different $f$-GANs and their minimum points.

$$\psi(\gamma) = \max_y \ \Psi(\gamma, y) \tag{91}$$

$$= \max_y \ \gamma y - (1 - \gamma)\,\hat{f}^*(y) \tag{92}$$

$$= (1 - \gamma) \max_y \ \frac{\gamma}{(1 - \gamma)} y - \hat{f}^*(y) \tag{93}$$

$$= (1 - \gamma)\hat{f}^{**}\left(\frac{\gamma}{1 - \gamma}\right) \tag{94}$$

$$= (1 - \gamma)\hat{f}\left(\frac{\gamma}{1 - \gamma}\right). \tag{95}$$

(Note that $\hat{f}^*(t) = \sup_{x \in dom_f}\{xt - f(x)\}$ by definition and $\hat{f}^{**} = \hat{f}$. Also, note that $\gamma \in [0, 1]$.)

We then examine whether these $f$-GANs fall into out theoretical framework. From Tables 6 and 7, we can see that many $f$-GAN instances satisfies the conditions in Theorems 4 and 5 and hence are considered well-behaved adversarial losses under our theoretical analysis. However, not all $f$-GANs satisfies Theorems 4 and 5. For example, the corresponding $\Psi$ functions for Kullback–Leibler (KL) and reverse KL divergences do not have a global minimum at $\gamma = \frac{1}{2}$.

## C   Graphs of the $\Psi$ and $\psi$ Functions for the $\omega$-weighted Losses

Figures 11 and 12 show the graphs of $\Psi$ and $\psi$, respectively, for the $\omega$-weighted versions of the classic, Wasserstein and hinge losses.

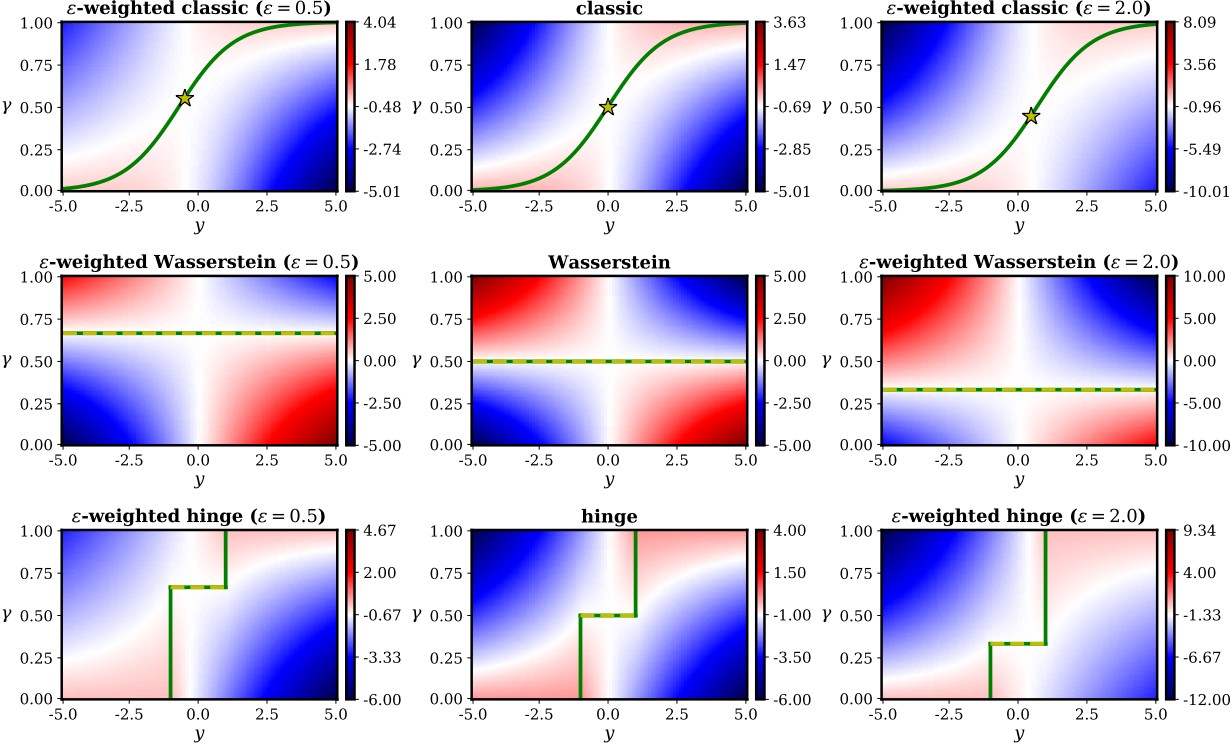

Figure 11: Graphs of the corresponding $\Psi$ functions for the $\omega$-weighted versions of the (top) classic and (middle) Wasserstein and (bottom) hinge losses (see Equation (20) for the definition of $\Psi$). The green lines show the domains of $\psi$, i.e., the values that $y$ can take for different $\gamma$. The stars and the yellow dashed lines indicate the global minima of $\psi$. The midpoints of the each color map is set to the minimum of $\psi$, i.e., the values taken at the star marks and yellow segments. Note that as $y \in \mathbb{R}$, we plot different portions of $y$ where the characteristics of $\Psi$ can be best observed. (Best viewed in color.)

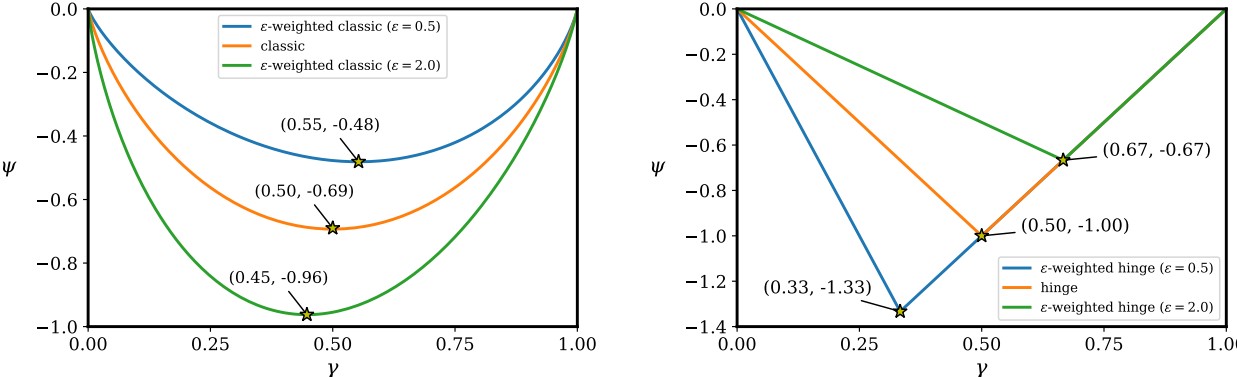

Figure 12: Graphs of the corresponding $\psi$ functions for the $\omega$-weighted versions of the (left) classic and (right) hinge losses (see Equation (21) for the definition of $\psi$). The stars indicate their minima. Note that for the Wasserstein loss, $\psi$ is only defined at $\gamma = \frac{2}{3}, \frac{1}{2}, \frac{1}{3}$ when $\omega = 0.5, 1.0, 2.0$, respectively, where it takes the value of zero, and thus we do not include the Wasserstein loss in this figure.

