# OpenReview forum: "On Output Activation Functions for Adversarial Losses: A Theoretical Analysis via Variational Divergence Minimization and An Empirical Study on MNIST Classification"
_TMLR — Rejected by TMLR_

### Review · Reviewer_km6w · 2022-08-13

**Summary Of Contributions:**

The paper proposes an analysis of the different adversarial loss function and regularization used for training GANs. Using the generalized adversarial loss proposed by Jolicoeur-Martineau (2019) they first provide conditions on the activation functions for the discriminator that leads to loss which are proper divergence. They show that the standard loss for GANs all satisfy these conditions, they also propose to new loss to illustrate the generality of the conditions.
In the second part of the paper, the authors conduct an extensive empirical comparison of the different losses and of different regularization commonly used to train GANs. The empirical results provide some insights into which loss and regularization seems to work better experimentally, the conclusion seems to be that although there is no clear winner, there is definitely some losses and regularization that performs worse, in particular it seems that the hinge loss combined with the local gradient penalty performs quite well.

**Broader Impact Concerns:**

Broader Impact statement is not necessary for this paper.

**Requested Changes:**

- Can the author provide a more details in section 6, why they propose this two new loss function. Why did they choose those particular two loss and not other loss that might also satisfy theorem 5 ?.
- Further experiments on a different dataset than MNIST, on GANs or on a different architecture, would be a valuable addition in order to see if results transfer to other settings. (It's not necessary to re-run all the combinations but could be interesting to just try a subset similar to some of the other experiments of the paper that look at some of the hyperparameters).

Minor comments and questions:
- The author mention the DAN abbreviation in the introduction of the paper without defining it.
- Typo in the last paragraph of section 2 Wasserstein instead of Wasserstein
- In the paper the author often mention Wasserstein but often when they do so they ignore the Lipschitz constraint. I understand the point of the author since they considered the unconstrained case and focus on the activation, but without the Lipschitz constraint the loss is not the Wasserstein distance anymore, it's some other distance. In particular the Wasserstein distance enjoys nice properties which are lost when we remove the Lipschitz constraint. This should be clarified a little bit in the paper.
- The author mention the relativistic loss form Jolicoeur-Martineau (2019) but don't define it, for completeness I believe it would be interesting that you define it in the paper.
- In Figure 3, you mention that $\psi$ is only defined for $\gamma=0.5$, does the analysis still holds in this case. Since $\psi$ is not well defined how do the necessary and sufficient conditions apply to Wasserstein ?
- Page 8 is missing the end of the last sentence of the page.
- The variance of Relativistic Hinge with TCGP is very high, I wonder whether this representative of this particular combination or resulted from "bad luck" and a single experiment completely failing and thus skewing the results.

**Strengths And Weaknesses:**

Strength:
- The paper is well written, while it's not novel it provides a nice presentation of adversarial losses under the framework of Jolicoeur-Martineau (2019), and explain clearly how the different popular loss for GANs fit into this framework.
- The theoretical analysis of the loss is new. The idea of analysing adversarial loss using $\Psi$ and $\psi$ is interesting, this provides interesting conditions for a loss to be a divergence and enable to explore and potentially invent new loss with properties that we want.
- The experiments are quite extensive were the authors compare a lot of different combinations of loss and regularization, as well as control for other hyperparameters such as momentum and the regularization coefficient.
- The paper is quite rigorous providing a good amount of details, as well as mentioning and discussing the limitations of the paper.

Weakness:
- The conclusion of the paper seem pretty limited or already known. Although the theoretical analysis is new it's not clear what insight it provides apart from being able to if a particular choice of activations leads to a divergence or not, which in my opinion is limited. It would have been nice to try to analyze what properties could actually make for a good loss. In particular the authors propose two new loss, which apart from being original don't seem to bring much, also the author don't provide an explanation about why we should consider those particular two loss and not other loss that might also satisfy the conditions.
- Although the experiment are extensive they're done on MNIST which is pretty small scale compared to current state of the art. The experiment are also done on DAN and not GAN which as the author rightfully mention is also a limitation of the work, and it would be interesting to see if the observations transfer to the GAN setting. I'm relatively confident that some observations would transfer, since the best performing loss seem to be the Hinge loss which has been used with success to train large scale GANs.
- The conclusion of the experimental results is again not very novel, and it's not very clear what to take from it. Overall it seems that the Hinge Loss and the Local Gradient Penalty works well, but this already kinda the standard setting for training GANs (actually it seems that recently SN has been more popular, so maybe it's worth trying to use GP again).
- I didn't understand what the author meant in section 6.6. Can the author try to clarify this section and what is the message of this section. From my understanding multiplying by a constant one side but not the other breaks the conditions of Theorem 5. However does the loss still verify the property 2 ?

---

> ### Author Response · Authors · 2022-09-08
> **Response to Reviewer km6w**
>
> Thank you very much for your valuable feedback! Ｗe have revised our manuscript according to your suggestions and comments. All the changes are highlighted in blue in the revised manuscript.
>
> For the requested changes, we have added some details in Section 6 regarding the reason for proposing the two new losses. In short, we proposed the absolute losses as it simplifies the quadratic functions to linear functions in least squares loss. For the asymmetric, we are interested in examining the possibilities of asymmetric f and g as all other adversarial losses considered in this paper have symmetric f and g.
>
> Regarding the purpose of Section 6.6, we are interested in examining what happens when Theorem 5 is violated. To be clear, it’s hard to verify whether Property 2 holds or not for a specific pair of functions f and g, and that’s why we derive Theorem 4 and 5 for easy verification. From the experimental results, we can see that violating the conditions in Theorem 5 does not necessarily lead to a complete failure, but rather a smoothly increasing error rate. This suggests that adversarial loss is not black and white – there are good and bad adversarial losses, where the bad ones are not complete failures and can still provide some supervision signal. This also relates to your comments on finding the properties of a good loss. And we believe that this would be the key to understanding adversarial losses in future study.
>
> Regarding the undefined $\psi$ problem, the theoretical analysis still holds in this case. For the Wasserstein loss, $\psi$ is undefined when $\gamma = 0.5$ as the maximum over $\Psi$ is unbounded (we added this information in the caption of Figure 3), and this does not break the proof. We added Footnote 5 for clarification.
>
> Regarding the high variance on the relativistic hinge with TCGP, it is true that one of the experiment runs is completely failing. We are aware of the instability and randomness of GAN training. However, we find that this phenomenon is less present in our case as the generator is trained to solve a relatively simple MNIST classification task, which can be verified by the relatively lower variance as compared to GAN evaluation results. To address this issue, we also added Figure 5 and 6 for further analysis. Specifically, we computed the maximum and minimum error rates achieved over ten runs to analyze the performance of an output activation function and its robustness when combined with different regularization approaches.
>
> We also made the following changes according to your suggestions.
> - We fixed the missing abbreviation definition in Section 1 and the typo in Section 2.
> - We added Footnote 8 to clarify that we will refer to the Wasserstein loss as the output activation function only.
> - We added Section 3.4 for introducing the relativistic loss for completeness.
> - We removed the misplaced incomplete sentence on page 8.

---

> ### Comment · Reviewer_km6w · 2022-09-14
> **Request for further experiments**
>
> Thanks for you response.
> I think the paper is quite interesting but I would really like if there was experiments with GANs, to see if the observations still holds on a more complicated task. Even if you report that the observations don't hold on other tasks, I would still find those observations highly valuable. Thus I really encourage you to consider running several experiments with GANs on either MNIST, CIFAR-10 or CelebA (I have a slight preference for CelebA and CIFAR-10, but if you have limited computing resources MNIST can be an option).

---

### Review · Reviewer_pS9e · 2022-08-15

**Summary Of Contributions:**

This paper studies the activation (after final discriminator output) and regularization of adversarial losses. Specifically, the paper has two contributions:
1. The necessary and sufficient conditions for the activations of adversarial losses.
2. Empirical studies on various combinations of activations and regularizations.

**Requested Changes:**

Despite the critical concerns on the empirical analysis, the theories on generalized adversarial loss and empirical observations may be helpful (at least it guides some suboptimal combinations for toy scenarios). I do not request to rerun all the experiments but suggest the authors organize current findings better and check if the key findings generalize to more realistic scenarios. Specifically,
- Clarify the connection and discussion between prior theories of GANs.
- Highlight the key observations from the empirical studies (e.g., R1/R2 reg, relativistic loss) in the introduction/contribution section.
- Although rerunning all the experiments is too costly, the authors could check if the key findings generalize to more realistic scenarios, e.g., GANs.

**Strengths And Weaknesses:**

Strength
- This paper is well written, clearly stating what this paper does and what the limitations are.
- The theoretical analysis of generalized adversarial loss is interesting.

Weakness
- How can the analysis of generalized adversarial loss be related to prior work? For example, can f-GAN and IPM-based GAN satisfy the condition in Theorem 5? It would be much more informative to add the discussion and connection with prior theories of GANs.
- As the paper also mentions, the biggest concern is whether the findings from DAN can be generalized to more realistic scenarios. For example, generating labels (instead of data) is somewhat artificial and different from the usual scenarios of GANs. The analysis could be much more informative if it were considered a more realistic setup, although providing fewer combinations.
- The adversarial loss can also be applied for representation (not the output), e.g., domain adaptation. Can the observation from DAN be generalized for these scenarios?
- The message from the empirical study is unclear. The paper says that there are no single wins. However, maybe because the task is too easy? Can this observation extend to more challenging tasks? Kurach et al. claim that gradient penalty and spectral normalization give consistent improvements. Table 4 of this paper seems to show similar things. Does this paper imply that R1 and R2 penalty (used in a popular architecture, StyleGAN) is a suboptimal choice? Highlighting key observations would be useful, although it is unclear if they can be generalized to other setups.

Editorial comments
- It is hard to understand what "activation" means in the first read. The definition comes in Section 4, but the paper could clarify that it implies the final activation after the discriminator output in front of the paper.
- The title is too broad and ambiguous. The paper could add more details about what the paper is doing.
- Typos: Last senctence in page 8, "penalties.s" in page 11.

---

> ### Author Response · Authors · 2022-09-08
> **Response to Reviewer pS9e**
>
> Thank you very much for your valuable feedback! We have revised our manuscript according to your suggestions and comments. All the changes are highlighted in blue in the revised manuscript.
>
> Regarding the highlights of the empirical analysis, we added Figure 5 and 6 for further analysis. Specifically, we computed the maximum and minimum error rates achieved over ten runs to analyze the performance of an output activation function and its robustness when combined with different regularization approaches. We also highlighted some key observations in the last paragraph of the introduction section (Section 1).
>
> Regarding the connections to prior theories of GANs, we added Appendix B for the connections to f-GANs. In short, f-GANs and our theoretical results have some overlaps, but both also reveal an exclusive class of adversarial losses. IPM-based GANs do not fall into our theoretical analysis as we consider the Lipschitz constraint as one of the regularization approaches. We added a sentence in Section 4.7 for clarity.
>
> We also made the following changes according to your suggestions.
> - We changed the term “activation function” to _“output activation function”_ to avoid confusion.
> - We updated the title of the paper to be more specific on our scope and contributions.
> - We fixed the typos on page 8 and 11.

---

> > ### Comment · Reviewer_pS9e · 2022-09-13
> > **Request for the Revision**
> >
> > Thank you for the rebuttal! It addressed most of my concerns except the empirical validation.
> >
> > After the discussion with other reviewers, I think this work is valuable to the community. However, the current empirical validation of DAN on MNIST is too limited.
> >
> > In the next round of revision, I ask the authors to provide the results on more realistic setups, e.g., **GAN on CIFAR-10 or CelebA**. Regardless of the previous observations from DAN on MNIST also holds for this setting, this new experiment will give valuable insights.

---

### Review · Reviewer_ZFng · 2022-08-29

**Summary Of Contributions:**

The paper provides a framework for studying adversarial losses. The authors put a wide range of previously studied adversarial loss functions into a common framework and propose a theoretical analysis to understand how components of adversarial loss functions interact. The authors also use their framework and analysis to propose new adversarial loss functions. Finally, the authors complement their theoretical framework with a broad range of experiments where they evaluate adversarial losses for image classification on MNIST.

**Broader Impact Concerns:**

The paper compares adversarial loss functions, which have been extensively studied in the literature over the past few years. There are some potential dangers associated with adversarial losses used for generative modeling, e.g., fake content generation. However, the paper provides no new practically relevant capabilities in this regard and hence raises no new ethical concerns from my perspective and does not require a Broader Impact Statement.

**Requested Changes:**

As mentioned above, my main concern is the experimental evaluation. I see two possible ways to address this:

A: Expand the limitations section 6.7. It may also be helpful for the reader to make the title more specific since currently it is rather broad and not commensurate with the limitations of the experiments.

B: Expand the experiments to involve generative tasks, ideally on more contemporary datasets.

If the paper appropriately scopes its claims, an updated version may be in line with the TMLR reviewing guidelines, specifically:

> Notably, the acceptance criteria for TMLR is technical correctness and clarity of presentation, rather than significance or impact. This is mostly assessed by determining whether the claims made in the submission are supported by accurate, convincing and clear evidence.
>
> An implication of the above is that, in a situation where some of the claims made by a submission aren't backed by sufficient evidence, instead of asking the authors to provide new evidence, one may instead ask that the authors reduce or adjust the claims made in the submission. Make sure to consider that possibility, as you discuss with the authors.

From my perspective, it is still unclear whether an experimental evaluation only on MNIST is sufficient for a publication in TMLR, even if the claims are properly scoped. I will leave this determination to the action editor assigned to this paper.

**Strengths And Weaknesses:**

**Strengths**

- The paper is well written.
- The paper surveys prior work well and puts prior adversarial loss functions into a coherent framework.
- The theoretical analysis in the paper is interesting and the authors use it to propose new adversarial loss functions.

**Weaknesses**

The main weakness is the experimental evaluation, specifically whether the experimental evaluation provides useful information for how adversarial losses are commonly used. The following points limit the external validity of the experiments:

- The experimental evaluation only compares the adversarial losses for a *classification* task via the DAN (discriminative adversarial networks) approach. This is an unusual choice because the adversarial losses studied in the paper have predominantly been used for *generative* modeling via GANs. It is not clear that the findings for classification tasks translate to the more common use case in generative modeling.

- The experiments involve only a single dataset, MNIST. I understand that a broad experimental comparison is computationally demanding, and that the authors may not have the resources to run experiments on larger datasets. Nevertheless, the large discrepancy between MNIST and current applications of GANs on more complicated image distributions unfortunately leaves open whether the findings transfer to more interesting use cases of GANs on larger datasets.

- Even within MNIST, the error rates seem high. Table 4 provides a convenient overview and shows that the best method in the suite of experiments achieves about 95% accuracy on MNIST. This is a comparatively low accuracy on MNIST since convolutional neural networks have achieved 99% accuracy on MNIST since 1998. Hence none of the DAN-based approaches studied in the experiments is competitive with current methods. Again, this leaves open the question whether the findings in the experiments provide guidance for the state of the art in image classification or generation.

Beyond the main points above, I have the following minor comments:
- Are the hyperparameter ranges broad enough? Specifically, Figure 7 b seems to indicate that an even smaller $\gamma$ may work better. Similarly, Figure 8 b suggests that increasing $\beta_1^{(D)}$ may work better.
- It could be useful to provide the total compute time the experiments took - a rough estimate of the overall total in Section 6.1 would suffice.
- I find the term "activation function" as defined just below Equation (9) confusing because the term "activation function" is commonly used for any neuron in a neural network, not just the output layer. What about introducing a new term, e.g., "output function" to refer to $f$, $g$, and $h$ specifically?
- The last sentence on Page 8 appears incomplete.
- Section 4.6 claims that the proposed losses have a lower computational cost. It would be helpful for the reader to be more precise about this claim, e.g., by quantifying the time complexity improvement or providing experimental data.
- The word "either" in the phrase "either the normalization layer" in Section 6.1 seems misplaced?

---

> ### Author Response · Authors · 2022-09-08
> **Response to Reviewer ZFng**
>
> Thank you very much for your valuable feedback! We have revised our manuscript according to your suggestions and comments. All the changes are highlighted in blue in the revised manuscript.
>
> Regarding the error rates, we have confirmed that the DAN approach considered in this paper is comparable in performance to a classifier with the same architecture trained with the cross entropy loss. Specifically, we trained the generator (a classifier in our case) using the cross entropy loss and observed an error rate of 4.12%. This is not far from the lowest error rates (4.71%) we achieved using the relativistic hinge loss and two-side local gradient penalty. As we are not aiming for state-of-the-art performance on MNIST classification in this work, we decided to go for a simple CNN architecture and focus on the relative performance in our comparative study. We have included this information in Footnote 9 in Section 6.2.
>
> Regarding the choice of hyperparameters, in the main comparative experiment, we followed the suggestions in the paper and used $\lambda=10$. According to Figures 7b and 8b, smaller $\lambda$ and larger $\beta^{D}_1$ does lower the error rates for the nonsaturating, Wasserstein and hinge losses. Hence, this remains as one of the limitations of our study. We have added a discussion on this in Section 6.7.
>
> We also made the following changes according to your suggestions.
> - We changed the term “activation function” to _"output activation function"_ to avoid confusion.
> - We removed the argument regarding computation cost for the two new losses as the output activation function is used only once and the saving is mostly neglectable.
> - We removed the misplaced incomplete sentence on page 8. We also removed the misplaced "either" in Section 6.1.
>
> For the requested changes, we have updated the title of the paper to be more specific on our scope and contributions. We also expanded Section 6.7 to discuss other limitations, including the lack of an extensive hyperparameter search.

---

### Decision · Action_Editors · 2022-10-13

**Recommendation:** Reject

**Comment:**

The paper provides a framework for studying adversarial losses. First, the authors provide the necessary and sufficient theoretical conditions for the activations of adversarial losses. Second, the authors complement their theoretical framework with a broad range of experiments where they evaluate adversarial losses for MNIST classification. All reviewers highlight new theoretical results in this paper. However, all agree that the empirical results on MNIST classification are too limited to provide any useful insights or findings, of interest for TMLR audience. All reviewers and AE also agree that if the authors consider GAN-based generation (instead of classification) in their experiments, then the paper would become much stronger and of interest for a broader range of audience (irrespectively whether the results are consistent with those for MNIST classification or not). AE also cannot find any meaningful or useful findings provided by the theoretical frameworks or empirical evaluations provided by the authors, e.g., as Reviewer km6w mentioned, "it's not clear what insight it provides apart from being able to if a particular choice of activations leads to a divergence or not." Overall, AE recommend rejection at the current form.

**Audience:**

It is quite arguable to say some individuals in TMLR's audience might be interested in knowing the findings of this paper. AE thinks the theoretical analysis provided by the authors might attract pure theorists interested in adversarial training. However, empirical results are mostly done for MNIST classification, which I doubt can attract much TMLR's audience.

**Claims And Evidence:**

The authors made two claims for (a) what types of output activation functions form a well-behaved adversarial loss? and (b) How different combinations of output activation functions and regularization approaches perform empirically against one another? For (a), the authors provide a new theoretical framework. For (b), the authors provide extensive experimental evidences under MNIST dataset and DAN (discriminative adversarial networks). Therefore, AE thinks the claims made in the submission are well supported by accurate, convincing and clear evidence.